# Assimilation of airborne gamma observations provides utility for snow estimation in forested environments

Eunsang Cho[1,2,3], Yonghwan Kwon[1,2,4*], Sujay V. Kumar[1], Carrie M. Vuyovich[1]

[1]Hydrological Sciences Laboratory, NASA Goddard Space Flight Center, Greenbelt, MD, USA
[2]Earth System Science Interdisciplinary Center, University of Maryland, College Park, MD, USA
[3]Now at Ingram School of Engineering, Texas State University, San Marcos, TX, USA
[4]Now at Korea Institute of Atmospheric Prediction Systems (KIAPS), Seoul, 07071, South Korea.

*Correspondence to*: Yonghwan Kwon (yhkwon@kiaps.org)

**Abstract.** An airborne gamma-ray remote sensing technique provides a strong potential to estimate reliable snow water equivalent (SWE) in forested environments where typical remote sensing techniques have large uncertainties. This study explores the utility of assimilating the temporally (up to four measurements during a winter period) and spatially sparse airborne gamma SWE observations into a land surface model (LSM) to improve SWE estimates in forested areas in the northeastern U.S. Here, we demonstrate that the airborne gamma SWE observations add value to the SWE estimates from the Noah LSM with multiple parameterization options (Noah-MP) via assimilation despite the limited number of measurements. Improvements are witnessed during the snow accumulation period while reduced skills are seen during the snow melting period. The efficacy of the gamma data is greater for areas with lower vegetation cover fraction and topographic heterogeneity ranges, and it is still effective in reducing the SWE estimation errors for areas with higher topographic heterogeneity. The gamma SWE data assimilation (DA) also shows a potential of extending the impact of flight line-based measurements to adjacent areas without observations by employing a localization approach. The localized DA reduces the modeled SWE estimation errors for adjacent grid cells up to 32-km distances from the flight lines. The enhanced performance of the gamma SWE DA is evident when the results are compared to those from assimilating the existing satellite-based SWE retrievals from the Advanced Microwave Scanning Radiometer 2 (AMSR2) for the same locations and time periods. Although there is still room for improvement, particularly for the melting period, this study shows that the gamma SWE DA is a promising method to improve the SWE estimates in forested areas.

## 1 Introduction

Seasonal snowpack is an important freshwater resource in snow-dominated regions, and thus accurate estimation of snow water equivalent (SWE) has been a pressing issue for managing water supply and forecasting snowmelt-driven flood events in a changing climate (Barnett et al., 2005; Cho et al., 2021; Musselman et al., 2021; Sturm et al., 2017). Due to its large variability, spatiotemporally continuous estimates of SWE cannot be generated by the existing in situ measurement network alone (e.g., Dozier, 2011). Large-scale distributions of SWE can be obtained from satellite remote sensing techniques such as passive microwave sensors (Derksen et al., 2005; Vuyovich et al., 2014); however, these are subject to errors resulting from retrieval algorithm limitations and uncertainties in certain conditions (Kang et al., 2014). Spatiotemporally continuous snow estimates at large scales can be generated by land surface modeling, which however suffer from large uncertainties associated with model physics, parameterizations, and meteorological boundary conditions (Broxton et al., 2016b; Cho et al., 2022; Kim et al., 2021; Raleigh et al., 2015; Yoon et al., 2019). Given the limitations of each method, data assimilation (DA) has been considered as a promising alternative to improve the SWE estimation skill as it systematically merges remote sensing observations with land surface model (LSM) predictions (e.g., Durand et al., 2009; Forman et al., 2012; Kwon et al., 2019; Liu et al., 2013; Zhang et al., 2014).

Given the sensitivity to snow properties and long record of observations, passive microwave brightness temperature ($T_B$) observations have been used to retrieve SWE or snow depth (e.g., Change et al., 1990; Derksen et al., 2010; Foster et al., 2005; Kelly et al., 2003; Kelly, 2009), and used within DA frameworks for the assimilation of $T_B$ (e.g., Durand and Margulis, 2006, 2007; Durand et al., 2009; Kwon et al., 2015, 2017; Larue et al., 2018a, 2018b) and $T_B$-based retrievals of SWE or snow depth (e.g., Dziubanski and Franz, 2016; Kumar et al., 2014). However, as mentioned above, $T_B$-based approaches are considered suboptimal for the following surface conditions: (1) deep snow, (2) wet snow, and (3) dense forest. Previous studies (e.g., Derksen et al., 2010; Kwon et al., 2019; Lemmetyinen et al., 2015) found that the $T_B$ signal, especially at high frequency (e.g., 36.5 GHz), saturates in deep snowpacks (i.e., when SWE is greater than 100 to 200 mm), which hampers microwave $T_B$-based SWE estimations. In the presence of wet snow, the $T_B$ sensitivity to SWE decreases because liquid water of snowpack dominates the $T_B$ signal due to the high emissivity of liquid water (Clifford, 2010; Walker and Goodison, 1993). Thus, the quality of the $T_B$-based SWE estimates is degraded under wet snow conditions (Kwon et al., 2019). The $T_B$ sensitivity to SWE also diminishes in forested areas (Roy et al., 2012) because the forest canopy blocks the microwave $T_B$ emission from the snowpack and emits its own $T_B$ signal (Foster et al., 1991), which adds considerable uncertainties in the $T_B$-based SWE estimates in forested areas (e.g., Kwon et al., 2016, Vuyovich et al., 2014). Vuyovich et al. (2014) showed specifically in the New England area that passive microwave retrievals underestimate SWE, though algorithms that account for forest fraction show improved performance. Although many enhancements have been proposed for the use of $T_B$ observations in estimating SWE, there are still significant limitations to overcome.

Recently, airborne remote sensing approaches such as Light detection and ranging (LiDAR) that have potential to overcome the existing challenges have been used within DA schemes to improve snow depth or SWE (e.g., Hedrick et al., 2018; Smyth

et al., 2019, 2020). Hedrick et al. (2018) focused on enhancing snow depth estimations over the Tuolumne River Basin in California by directly inserting the NASA Airborne Snow Observatory (ASO) LiDAR snow data into the iSnobal model (Mark et al., 1999). They found that agreement between the LiDAR snow depth and updated modeled snow depth was improved as compared to original modeled snow depth. Smyth et al. (2020) attempted to assimilate the ASO LiDAR snow depth observations into the Flexible Snow Model to improve snow density and SWE estimations. They showed that DA reduced snow density bias by over 40% and SWE bias by over 70% across eight climate zones in the western U.S. and in both wet and dry years. However, the impacts of known limitations such as forest cover and wet snow (in melting period) within a DA framework have not been widely examined, which were emphasized to be conducted in future research. Furthermore, most previous studies have mainly focused on the western U.S. environments (e.g. mountainous regions) with limited investigations in other regions such as temperate forest environments over northeastern U.S.

As a historically well-established remote sensing technique, the airborne gamma radiation technique provides an opportunity to estimate reliable SWE, because the gamma approach uses the attenuation of the terrestrial gamma-ray emission by water in the snowpack (any phase) with minimal effects by wet snow and dense forest (Carroll, 2001; Carroll and Vose, 1984; Goodison et al., 1984). Since the early 1980s, airborne gamma radiation snow surveys operated by the National Oceanic and Atmospheric Administration's (NOAA) Office of Water Prediction (OWP; formerly by the National Operational Hydrologic Remote Sensing Center [NOHRSC]) have provided SWE observations to regional NOAA National Weather Service (NWS) River Forecast Centers (RFCs) and other agencies across the United States and southern Canada to support operational flood forecasting system and water supply outlooks (Carroll, 2001; Peck et al., 1980). Recently, Cho et al. (2020b) found that the long-term gamma SWE observations have a remarkable agreement with ground-based gridded SWE products particularly in forest regions (R-value = 0.73 and 0.72 and Bias = 0.0 and -1.3 cm for mixed forest and deciduous forest, respectively), implying that the gamma-based SWE observations have the potential to be used in a DA framework to improve modeled SWE estimates. While the airborne gamma SWE products along with in-situ snow depth and SWE and satellite-based snow cover areas are currently assimilated into the NWS SNOw Data Assimilation System (SNODAS) to provide the near-real-time, high spatial resolution (1 km$^2$ gridded) SWE information (Barrett, 2003), how much the gamma radiation SWE retrievals help improve the modeled SWE estimates is not well quantified particularly in a forested region.

The objective of this study is to evaluate the potential of the airborne gamma SWE retrievals within a DA framework to enhance SWE estimates in a temperate forest environment in the northeastern U.S. More specifically, we aim to answer three research questions: (1) How much is the modeled SWE improved by assimilating the airborne gamma SWE into a model? (2) Do land surface characteristics such as forest density, slope, and elevation affect the assimilation performance? (3) Can the spatial sparseness of the gamma SWE observations be overcome by employing the localized DA approach? In this study, the Noah LSM with multi-parameterization options (Noah-MP) is used to assimilate the long-term airborne gamma radiation SWE observations with the ensemble Kalman filter (EnKF) scheme within the NASA Land Information System (LIS). This paper is organized as follows. Section 2 provides the study area with general land cover characteristics. Section 3 describes the datasets including the airborne gamma radiation survey, reference SWE data, tree cover fraction, and topographic feature

variables. The description of the Noah-MP model with assimilation scheme is included in the section 4. Section 5 presents evaluation results of DA SWE performances with discussion about the similarities, differences, and new findings in the results with respect to previous studies. Conclusion and future perspectives are drawn in section 6.

## 2 Study Area

The study area comprises parts of the northeastern United States, including New Hampshire and Maine with heavily forested regions which remain a challenging region in snow remote sensing and modeling communities. The dominant seasonal snow class in this region is montane forest (**Figure 1a**; Sturm and Liston, 2021). Land cover types are mainly deciduous broadleaf forest and mixed forest. Fractional tree cover over the study area ranges from 70 to 100% based on the vegetation continuous field (VCF) map from the NASA Making Earth System Data Records for Use in Research Environments (Hansen and Song, 2018; **Figure 1b**). The NOAA OWP airborne gamma snow surveys occur almost every year over the designated flight lines (yellow lines in Figure 1b).

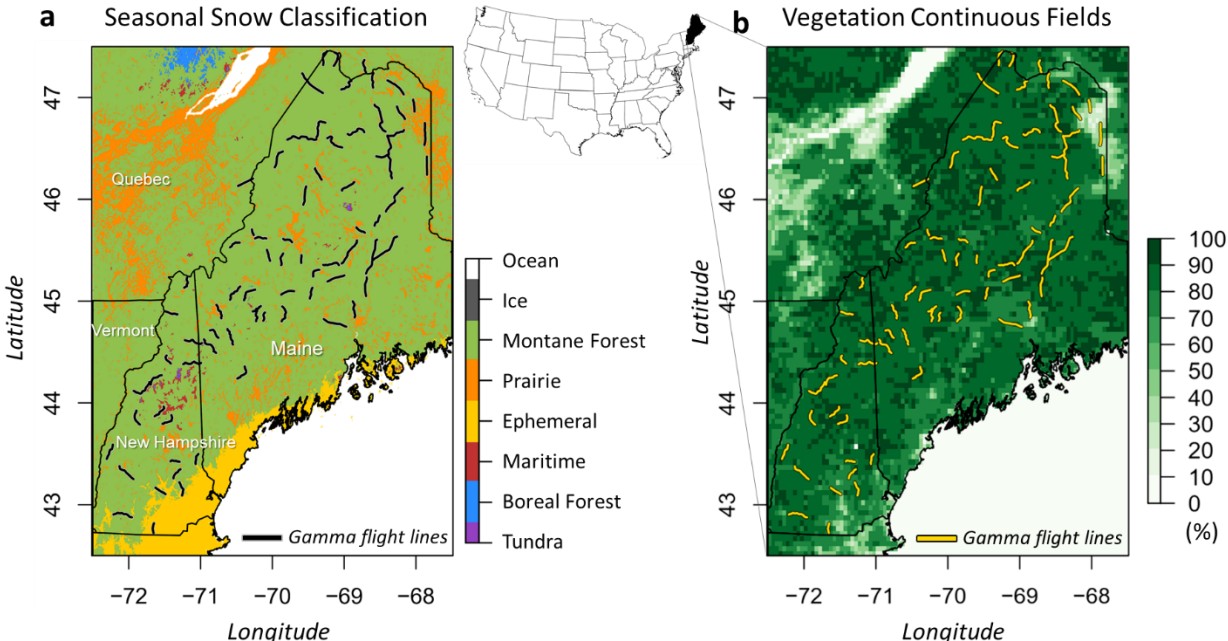

**Figure 1**. (a) Sturm and Liston's new seasonal snow classification (Sturm and Liston, 2021), and (b) Vegetation Continuous Field maps of the study area over the northeastern United States with the NOAA airborne gamma flight lines

## 3 Data

### 3.1 NOAA airborne gamma snow survey

The operational airborne gamma radiation snow and soil moisture survey operated by the NOAA's OWP has been conducted to observe near real-time areal SWE (Carroll, 2001) throughout the United States and Canada provinces since 1979. The gamma SWE observations have been used by the NWS Hydrologic Services Program for spring flood forecasts and water supply outlook. The key principle of the gamma SWE technique is the attenuation of the natural gamma-ray signal due to the snowpack (Carroll, 2001; Peck et al., 1980). The gamma SWE values are estimated using the difference in the rates of gamma radioisotopes ($^{40}K_0$, $^{208}Tl_0$, and gross count, $GC_0$) between over bare and snow-covered land surface (Cho et al., 2020a). The gamma-ray signal for designated flight lines are measured in the fall prior to freezing onset and then revisited in the winter. A gamma radiation detector equipped on a low-flying aircraft observes the gamma-ray particles. This detector measures terrestrial gamma radiation naturally emitted from trace elements of the three radioisotopes in the upper 20 cm of soil. The operational approach assumes the gamma rates over bare ground from the fall survey remain constant during the winter surveys. A typical gamma flight footprint covers approximately 5 km$^2$ (a 300 m wide and 16 km long). The final gamma SWE value is generated as an area-mean value for each flight path. The airborne gamma SWE values are estimated using the equations below:

$$SWE\left(^{40}K\right) = \frac{1}{A} \cdot \left[ ln\left(\frac{40K_b}{40K_s}\right) - ln\left(\frac{100 + 1.11 \cdot SM(40K_s)}{100 + 1.11 \cdot SM(40K_b)}\right)\right] \tag{1}$$

$$SWE\left(^{208}Tl\right) = \frac{1}{A} \cdot \left[ ln\left(\frac{208Tl_b}{208Tl_s}\right) - ln\left(\frac{100 + 1.11 \cdot SM(208Tl_s)}{100 + 1.11 \cdot SM(208Tl_b)}\right)\right] \tag{2}$$

$$SWE(GC) = \frac{1}{A} \cdot \left[ ln\left(\frac{GC_b}{GC_s}\right) - ln\left(\frac{100 + 1.11 \cdot SM(GC_s)}{100 + 1.11 \cdot SM(GC_b)}\right)\right] \tag{3}$$

where $^{40}K$, $^{208}Tl$, and $GC$ are uncollided gamma count rates in the top 20 cm of soil, and $SM(40K\ )$, $SM(208Tl\ )$, and $SM(GC\ )$ are the corresponding soil moisture values by weight (%). Subscripts $b$ and $s$ indicate bare and snow-covered grounds, respectively. The airborne gamma SWE ($SWE_{gamma}$; g cm$^{-2}$) is a weighted value by multiplying the three independent SWE estimates by weighting coefficients, 0.346, 0.518, and 0.136, and summing the calculated three values as below (Carroll, 2001; Jones and Carroll, 1983).

$$SWE_{gamma} = 0.346 \cdot SWE\left(^{40}K\right) + 0.518 \cdot SWE\left(^{208}Tl\right) + 0.136 \cdot SWE(GC) \tag{4}$$

The final SWE value is reported in the Standard Hydrometeorological Exchange Format (SHEF) product through the NOHRSC website (https://www.nohrsc.noaa.gov/snowsurvey/) (Carroll, 2001). In this study for dense forest environments, 1,508

airborne gamma SWE observations covering 79 flight lines flown over densely forested environments in the northeastern United States are used from January 1985 to May 2017.

### 3.2 UA SWE

The University of Arizona (UA) SWE is the ground observation-based 4-km gridded SWE product developed by consistently assimilating the snow telemetry (SNOTEL) SWE and NWS Cooperative Observer Program (COOP) snow depth measurements (which was first converted to SWE using a newly developed snow density parameterization) with the Parameter-elevation Regressions on Independent Slopes Model (PRISM) temperature and precipitation data over the continental United States (Broxton et al., 2016a; Dawson et al., 2017; Zeng et al., 2018). In this study, the UA SWE is used as reference data to evaluate and compare the open-loop and assimilation results from the Noah-MP simulations. The accuracy and robustness of the UA SWE product have been proven by examinations of point-to-point and pixel-to-pixel interpolations (Broxton et al., 2016a, b), and evaluations against independent ASO LiDAR-based SWE and gamma radiation SWE measurements (Cho et al., 2020b; Dawson et al., 2018). Cho et al. (2020b) demonstrated that the UA SWE product strongly agreed with the airborne gamma SWE regardless of land cover type and snow classification over the continental U.S. This product has been used as a reference SWE for multiple purposes such as quantifying uncertainties in land surface modeled SWE (Kim et al., 2020; Zhang et al., 2022); characterizing extreme events (Welty and Zeng, 2021), and estimating extreme values for infrastructure design (Cho and Jacobs, 2020). The daily UA SWE product (version 1) from October 1984 to December 2017 is used in this study, which is publicly available from the National Snow and Ice Data Center website (https://nsidc.org/data/nsidc-0719).

### 3.3 AMSR2 Passive Microwave SWE

For comparison purposes, the existing satellite-based SWE retrievals from the Advanced Microwave Scanning Radiometer 2 (AMSR2) were also assimilated in this study. AMSR2 passive microwave sensor is the follow-on instrument to the Advanced Microwave Scanning Radiometer for Earth Observing System on board Aqua satellite (AMSR-E; Imaoka et al., 2010). AMSR2 on board the Global Change Observation Mission - Water (GCOM-W1) satellite has measured daily scans at 1:30 a.m./p.m. local time at 1–2 days revisit frequency since May 2012. AMSR2 SWE product is calculated by using snow depth estimated from an empirical relationship between snow depth and $T_B$ observations at 18.7 and 36.5 GHz along with higher and lower frequencies and snow density values for each snow class from the Sturm's snow classification system (Kelly, 2009; Sturm et al., 2010). The Level 3 AMSR2 SWE products with the 10 km spatial grid were obtained from the JAXA GCOM-W1 Data providing service (http://gcom-w1.jaxa.jp). In this study, the AMSR2 data at descending overpass (01:30 a.m.) was used only to minimize the wet snow effect.

### 3.4 Tree cover fraction and topographic features

In this study, we used tree cover fraction (TCF) and topographic feature data sets to compare DA performance by the degrees of them. The NASA Making Earth System Data Records for Use in Research Environments (MEaSUREs) Vegetation

Continuous Fields (VCF5KYR; Version 1) provides annual global fractional vegetation cover maps with three layers including percent tree cover, percent bare ground, and percent non-tree vegetation at 0.05-degree spatial resolution from 1982 to 2016 (Hansen and Song, 2018). Among them, the percent tree cover was used. To account for the interannual variations in the fractional tree cover, annual TCF values were obtained for each gamma line. The elevation data (0.0083-degree grid) used in this study were an aggregated map using the Shuttle Radar Topography Mission (SRTM) 90 m resolution elevation data (Farr et al., 2007). The slope and elevation range maps with the same spatial grid were obtained using the "raster" R-package ("terrain" function in this package; Wilson et al., 2007). The elevation range, referred to as "topographic heterogeneity" in this manuscript, was calculated as the difference between the minimum and maximum elevation value among a given grid and its surrounding eight grids (total nine grids). The three topographic features were computed by areal-weighted average for each gamma flight footprint.

## 4. Model and Methods

### 4.1 Noah-MP

Noah-MP (v3.6; Niu et al., 2011; Yang et al., 2011) was employed to simulate snow variables such as SWE and snow depth. Noah-MP was developed based on the original Noah LSM (Ek et al., 2003) with improved representations of biophysical and hydrological processes. A grid cell in Noah-MP consists of one vegetation canopy layer, up to three layers (depending on the whole snow depth) of snowpack, four soil layers (with thicknesses of 0.1 m, 0.3 m, 0.6 m, and 1.0 m from top to bottom), and an unconfined aquifer layer. Regarding snow processes, intercepted snow exists in Noah-MP as solid and liquid phases on the vegetation canopy, and melting/refreezing of intercepted snow, dew/evaporation, and frost/sublimation on the vegetation canopy are explicitly represented in the model. Snow depth and SWE are simulated by considering snow layer compaction by the weight of the overlying snow layers, snow metamorphisms (destructive and melt), and snowmelt-refreeze processes. Physical parameterization scheme options used in the current study are listed below: (1) dynamic vegetation for the vegetation option; (2) Noah-type soil moisture factor for stomatal resistance (Chen and Dudhia, 2001); (3) Ball-Berry canopy stomatal resistance scheme (Ball et al., 1987); (4) TOPMODEL-based runoff scheme; (5) simple groundwater scheme (SIMGM; Niu et al., 2007); (6) general Monin-Obukhov similarity theory (M-O; Brutsaert, 1982) for surface layer drag coefficient; (7) NY06 scheme (Niu and Yang, 2006) for supercooled liquid water (or ice fraction) in frozen soil; (8) NY06 scheme (Niu and Yang, 2006) for frozen soil permeability; (9) modified two-stream radiation transfer scheme (Yang and Friedl, 2003, Niu and Yang, 2004); (10) Biosphere-Atmosphere Transfer Scheme (BATS) for the snow albedo (Yang and Dickinson, 1996); (11) Jordan91 scheme (Jordan, 1991) for partitioning precipitation into rainfall and snowfall; (12) original Noah scheme for lower boundary condition of soil temperature; and (13) semi-implicit snow and soil temperature time scheme. An ensemble of model initial conditions was constructed through a two-step spin-up procedure. First, a single-member model simulation was run for 40 years, from 1 January 1980 to 1 January 2020, driven by NASA Modern-Era Retrospective analysis for Research and Applications, version 2 (MERRA-2; Bosilovich et al., 2015) forcing. Then, using a restart file generated in the first step, an

additional 3-year spin-up, from 1 January 1981 to 1 March 1984, was conducted using 20 ensemble members to generate

model uncertainty metrics for the DA. The open-loop (OL; without assimilation) and DA experiments were run from 1 March 1984 to 1 October 2017 using the 20-member ensemble initial conditions. A model simulation time-step of 15 minutes was used, and daily mean outputs were evaluated.

## 4.2 Assimilation Scheme

Data assimilation experiments were conducted within the NASA LIS (Kumar et al., 2006; Kumar et al., 2008; Peters-Lidard

et al., 2007). The ensemble Kalman filter (EnKF) scheme was applied (Reichle et al., 2002) to assimilate airborne gamma radiation-based SWE retrievals into Noah-MP. In the EnKF scheme, model uncertainty is implicitly represented by the ensemble spread and an ensemble size of 20 was used in this study. The ensemble spread was generated by perturbing meteorological forcing fields and prognostic model state variables with the assumption of a Gaussian distribution. Perturbation parameters applied during the OL and DA runs are presented in Table 1, which are suggested by Kwon et al. (2021) based on

Forman et al. (2012), Kumar et al. (2009, 2014, 2016), and Reichle et al. (2008). When observations (i.e., airborne gamma SWE) are available, EnKF updates forecasted model state variables using the following equation:

$$M_i^+ = M_i^- + K(Obs - HM_i^-) \tag{5}$$

where $M_i^+$ is the updated (after assimilation) model states (i.e., SWE); $M_i^-$ is the forecasted (before assimilation) model states (i.e., SWE); $Obs$ is the gamma SWE retrievals; $H$ is the observation operator ($H = 1$ in this study); $i$ denotes the ensemble

member; and $K$ is the Kalman gain given by:

$$K = Cov(M_i^-, HM_i^-)\{Cov(HM_i^-, HM_i^-) + R\}^{-1} \tag{6}$$

where $Cov(M_i^-, HM_i^-) = Cov(HM_i^-, HM_i^-)$ is the covariance of the model forecasted SWE, and R is the covariance of the observation error. The gamma SWE retrieval error standard deviation of 23 mm was assumed based on realistic error values from previous studies such as Carroll and Vose (1984). Note that assimilation of gamma SWE updates only modeled SWE

and the amount of the SWE update is added to ice content of the bottom snow layer. Then, snow layer variables such as thickness, snow ice and liquid water content, and SWE of each snow layer are adjusted using the same methods as used in the Noah-MP's snow layer compaction, combination, and subdivision procedures.

**Table 1**. Perturbation parameters applied to model prognostic state variables and atmospheric forcing fields during the OL and DA runs.

| Variable | Perturbation Types | Std dev | AR(1) | Cross correlations | |
|---|---|---|---|---|---|
| | | | | *SWE* | *SD* |
| Model prognostic state variables | | | | | |
| *SWE* | M | 0.01 | 3 hr | – | 0.9 |
| Snow depth (*SD*) | M | 0.02 | 3 hr | 0.9 | – |

| Atmospheric forcing fields | | | | *SW* | *LW* | *P* |
|---|---|---|---|---|---|---|
| Shortwave radiation (*SW*) | M | 0.3 | 1 day | – | -0.5 | -0.8 |
| Longwave radiation (*LW*) | A | 50 W m$^{-2}$ | 1 day | -0.5 | – | 0.5 |
| Precipitation (*P*) | M | 0.5 | 1 day | -0.8 | 0.5 | – |

M: multiplicative; A: additive; AR(1): first-order autoregressive temporal correlation.

## 4.3 DA localization

Due to its sparsity in space, the airborne gamma radiation-based SWE observations can be limited to be used within the DA system. To quantify if the spatially sparse gamma SWE observations can improve the SWE estimates in the surrounding areas, where the observations are not available, we apply a distance-based localization method into the assimilation procedure. The localization is applied in the assimilation by weighting distances from the flight lines (up to a specified localization distance; *r*) using the Gaussian decay-based localization method as follows:

$$W = exp\left\{\frac{-d^2}{2 \cdot \left(\frac{r}{2}\right)^2}\right\} \tag{7}$$

where *d* is the distance between the updated grid cells (i.e., flight lines) and grid cells without observations within a specified localization radius *r*. The magnitude of the SWE DA adjustment for a grid cell from the assimilation is calculated using the localization weight (*W*) which is calculated based on the distance (*d*) from the updated grid cells overlapped with the flight line. If a grid cell is affected by multiple flight lines, an average of the updates is added to the prior SWE estimates of the grid cell. We apply a localization function with six different distances (e.g., 4, 8, 16, 24, 32, and 48 km from the lines). For an evaluation of the DA SWE with a given localization weight, the areal mean DA SWE time series are obtained for an effective

area buffered by a specified distance around the gamma flight line. The areal mean OL and UA SWE time series are also obtained in the same way to compare with the corresponding OL and UA SWE values.

## 5. Results and discussion

### 5.1 Comparison between DA and OL SWE with airborne gamma SWE

To examine the updated SWE performance over the gamma lines by assimilating airborne gamma observations into Noah-MP, statistical metrics were compared between OL and DA SWE using UA SWE (**Figure 2**). The values of 1:1 slope were closer to 1 (median slopes of OL and DA were 1.45 and 0.91, respectively) and RMSD values decreased by DA, even though negative biases were found. The absolute SWE bias was higher in the DA as compared to the OL simulation (**Figure 2**). However, this was a consequence of the fact that in correcting the overestimated SWE during the accumulation season, the DA introduced a greater underestimate during the melt season (**Figures 3** and **4**). The OL SWE was largely deviated from the 1:1 linear relationship during the snow accumulation season (i.e., January, February, and March) and early in the snowmelt season (i.e., April). **Figure 3** shows that the deviation was significantly reduced through the assimilation of the gamma SWE retrievals even though a reduced R-value was obtained.

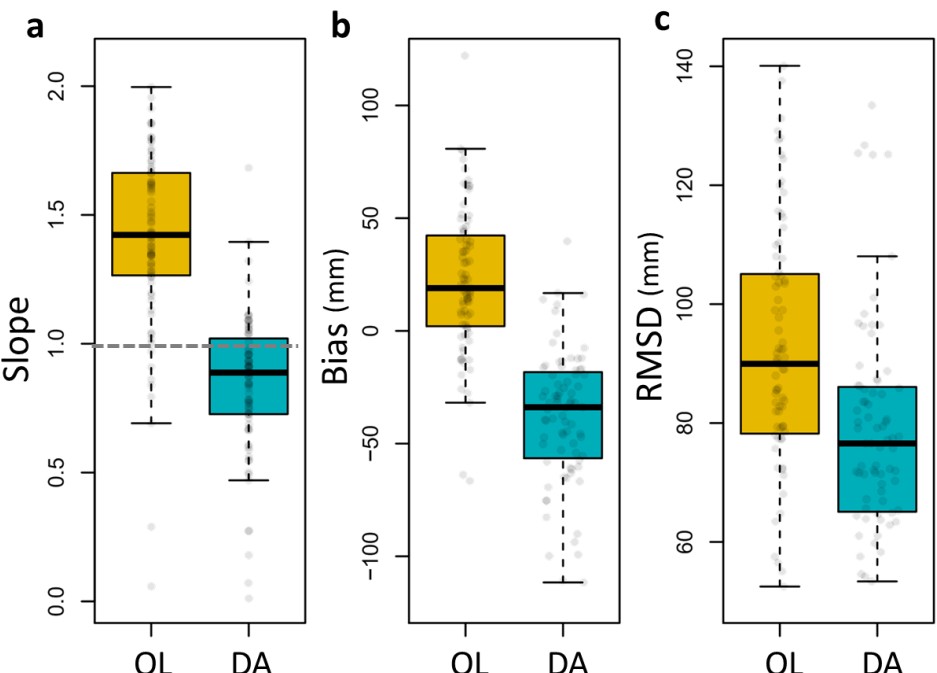

**Figure 2**. Comparison of statistics between open-loop (OL) SWE and data assimilated (DA) Noah-MP SWE estimates by using airborne gamma radiation SWE observations with the University of Arizona SWE from 1985 to 2017: (a) slope from 1:1 plot, (b) Bias, and (c) RMSD from a linear relationship between the estimated SWE and UA SWE.

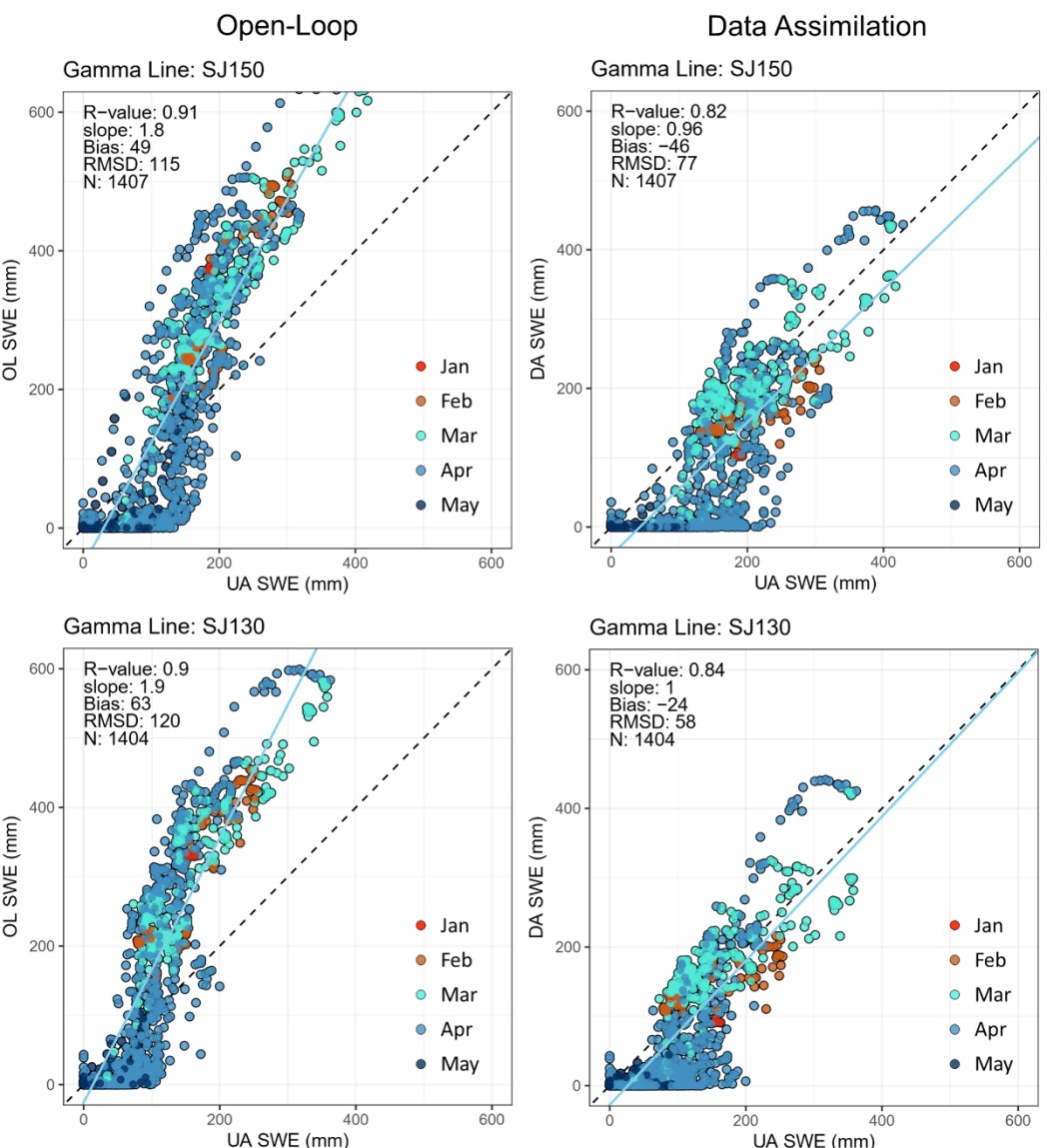

**Figure 3**. Examples of scatterplots of two gamma flight lines (SJ150 and SJ203) between the Noah-MP SWE estimates (from the open-loop (OL) and data assimilation (DA) experiments; y-axis) and daily University of Arizona SWE (x-axis) from October 1985 to May 2017 (total 33 water years). R-value, slope, Bias, RMSD, and number of data points (N) in the linear relationship are presented in the figures.

A promising aspect is that the assimilation of the temporally sparse (i.e., only one or two data points at the end of the snow accumulation period and/or early in the snowmelt period) airborne gamma SWE retrievals improved the model estimates of SWE, which was particularly noticeable in some lines and years, such as the gamma line SJ150 in WY1991 (**Figure 4**). For comparison purposes, results of assimilating the AMSR2 SWE retrievals were also plotted (green solid line in **Figure 4**). The

AMSR2 SWE was largely deviated (underestimated) from the UA SWE in densely forested areas, and assimilating the AMSR2 SWE data led to degradation of the SWE estimates. This further emphasizes the effectiveness of the gamma SWE data in improving the model estimates of SWE via assimilation in forested areas even with fewer available data compared to the AMSR2 SWE. However, the assimilation of the airborne gamma SWE measurements was not able to improve the snow ablation timing due to sparse gamma data during the spring in combination with the overall poor model performance during the melt season. As shown in **Figure 4**, compared to the UA SWE, Noah-MP simulated earlier snow melt-out despite the overestimated snow accumulation, which may be attributed to the Noah-MP model structure and physics (e.g., simplified representations of snow layers), parameterization schemes, and/or atmospheric forcing. Also, the peak SWE cannot be corrected if a single gamma SWE flight exists only after the accumulation period. The availability of more frequent gamma observations during both the accumulation and melt seasons could lead to further improvements in estimating SWE in the ablation period while the model and forcings need to be enhanced.

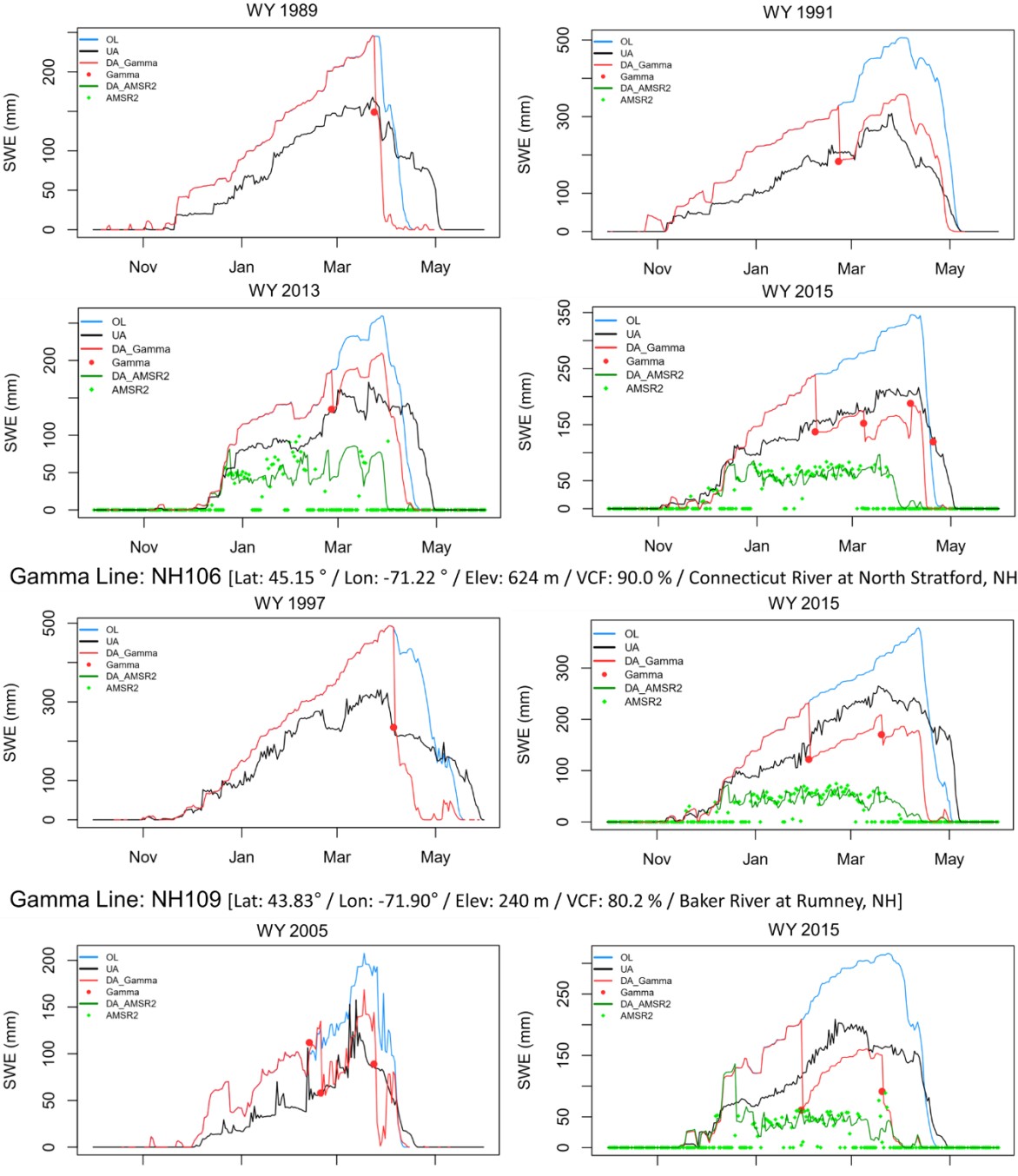

**Figure 4**. Examples of daily SWE time series of three gamma lines (SJ150, NH106, and NH109) with latitude (Lat), longitude (Lon), elevation (Elev), and vegetation cover fraction (VCF) for individual years including the open-loop (OL) and gamma data assimilated (DA_Gamma) Noah-MP SWE estimates along with the passive microwave SWE data from the Advanced Microwave Scanning Radiometer 2 (AMSR2) and AMSR2 data assimilated SWE (DA AMSR2).

**5.2 Effect of land surface characteristics on assimilation performance**

To examine effects of land surface characteristics on the DA performances as compared to the OL, the performance of the gamma SWE DA, presented as differences (i.e., DA minus OL) in the linear regression slope, bias, and RMSD, with the UA SWE were compared by four physical features, TCF, slope, elevation range (i.e., topographic heterogeneity), and elevation (**Figure 5**). Two groups of each land surface characteristics were determined by dividing the gamma flight lines into two (i.e., low and high) groups of equal numbers of the flight lines. For TCF, DA SWE in a group with low TCF (less than 85%) has lower bias and lower RMSD than OL SWE, while the DA led to a marginal improvement in the high TCF. Considering that the TCF values in the low group ranges from 31% to 84% (mean: 62 %), DA using airborne gamma SWE improved SWE over densely forested regions.

Differences in the DA performance between the low and high groups were observed for all surface characteristics. The 1:1 slope was improved by DA for both the low and high ranges of all surface characteristics. DA led to larger improvements in the 1:1 slope and RMSD for low VCF, slope, elevation range, and elevation. With respect to the bias, assimilation of the gamma SWE retrievals improved the group-averaged performance for both the low and high groups of the surface characteristics with larger improvement in low VCF and high slope, elevation range, and elevation. For individual physical characteristics, the added value of the gamma SWE data on the model SWE estimates via assimilation was greater for the low VCF range based on both bias and RMSD. It is worth noting that the low VCF ranges from 31% to 84%, and DA significantly improved the SWE, even for the high VCF (i.e., greater than 85%). This implies that the gamma-based SWE estimates within DA frameworks can be a promising alternative to traditional $T_B$-based approaches in forested areas. Comparable DA performance patterns were also obtained for other land surface characteristics. Although the gamma SWE DA exhibited smaller RMSD improvements in areas with higher topographic heterogeneity, than those with lower ranges, it was still effective in reducing error statistics.

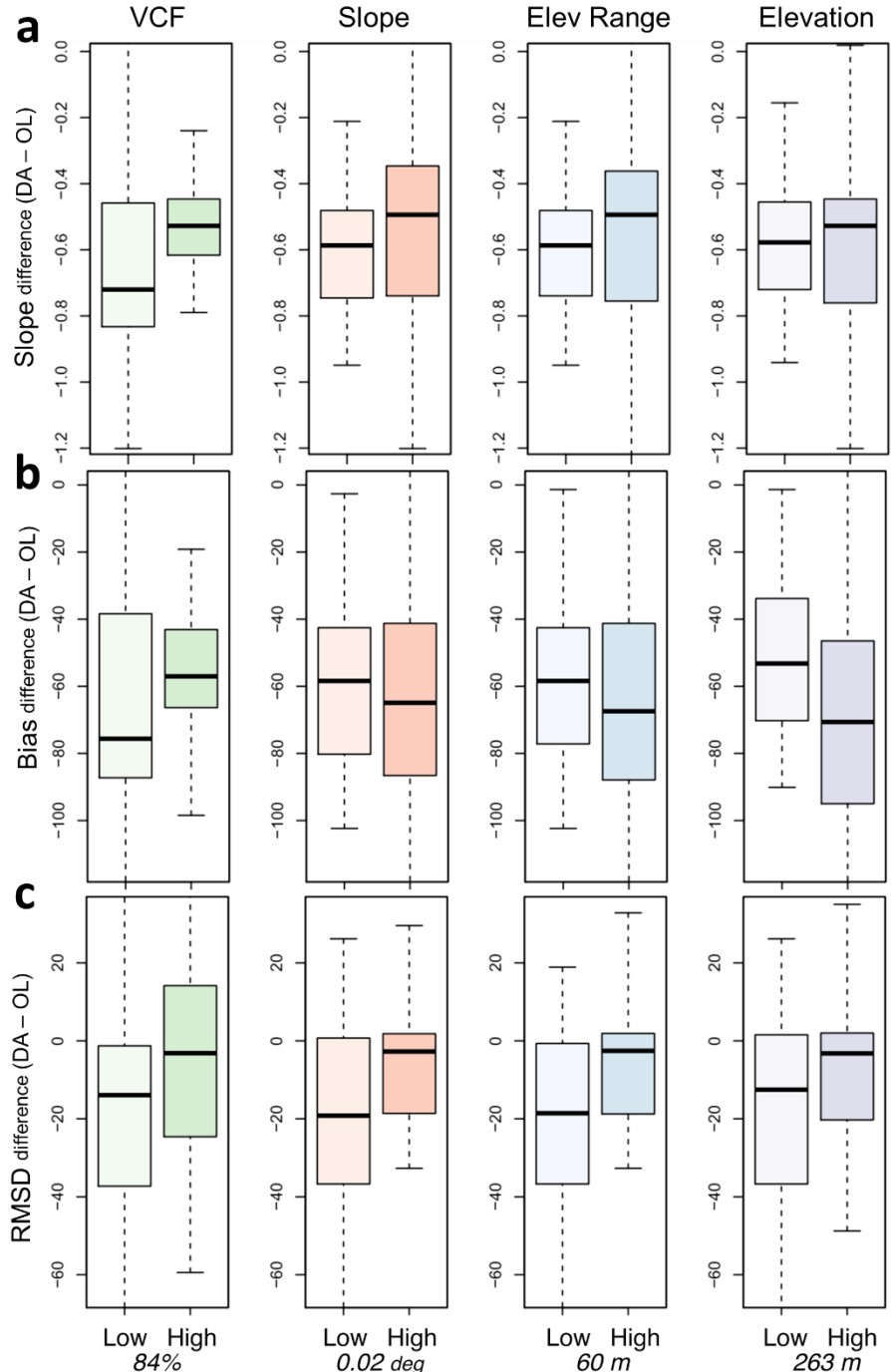

**Figure 5**. Boxplots of differences in (a) Slope from 1:1 plot, (b) Bias, and (c) RMSD between the DA and OL cases (computed as DA – OL) with respect to vegetation cover fraction (VCF), slope (degree), elevation range (m), and elevation (m). The two groups (low/high) were divided into equal numbers of values. The bottom values are 50% quantile values for each characteristic.

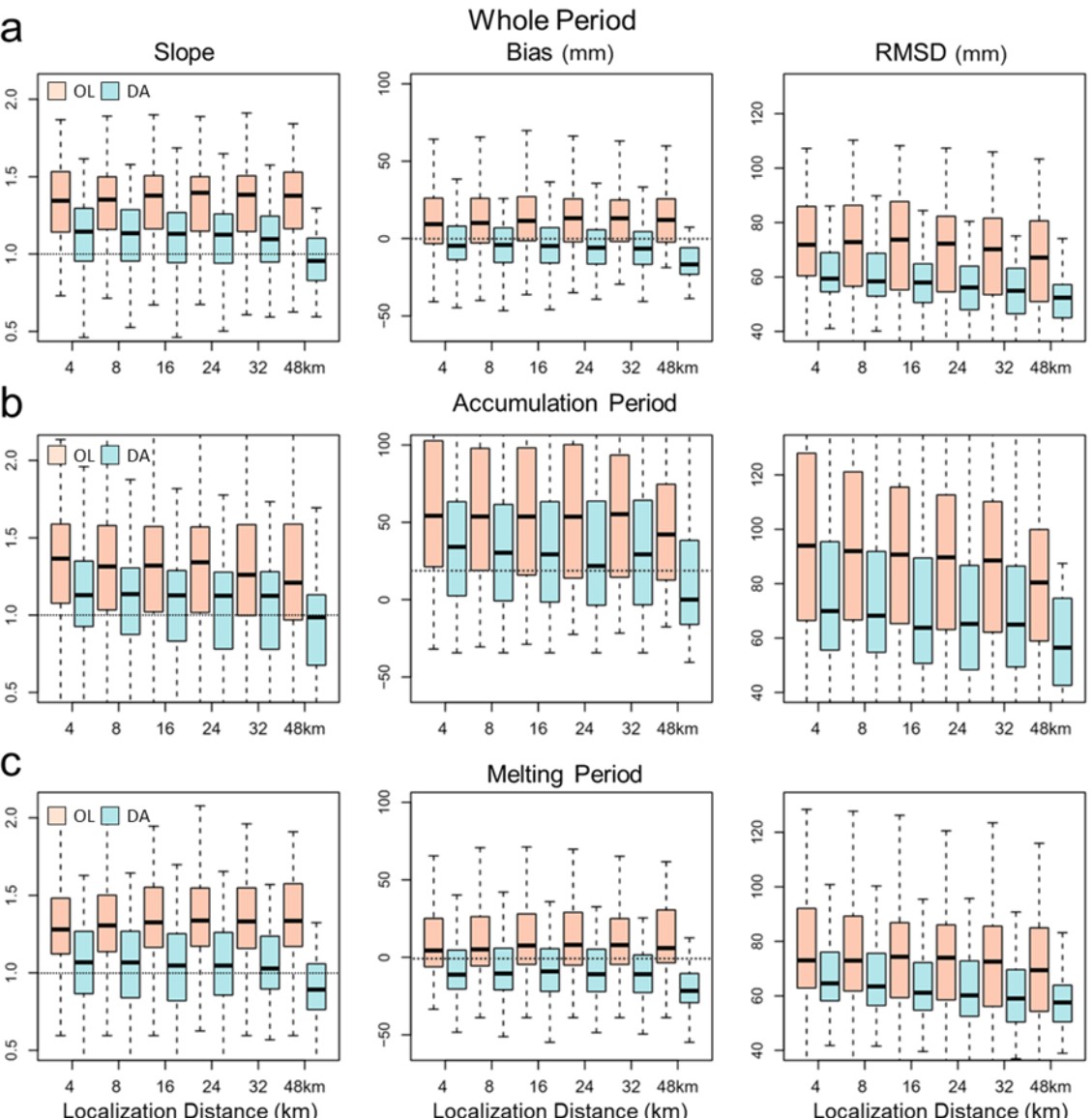

**Figure 6**. Localized data assimilation (DA) and open-loop (OL) Noah-MP SWE performances as compared to the UA SWE with different localization distances (e.g., 4, 8, 16, 24, 32, and 48 km) for the whole (accumulation and melting periods), accumulation, and melting periods, respectively.

One of the limitations of the airborne gamma SWE observations is a limited spatial coverage, which is typically 5-7 km$^2$ with a swath 300 m wide and 15-30 km long. It is necessary to assess if the spatially sparse airborne gamma SWE observations can also improve the SWE estimates in areas surrounding the gamma flights via assimilation. Here, the DA experimental cases that employ a localization function with different distances (e.g., 4, 8, 16, 24, 32, and 48 km from the flight lines) are evaluated

(**Figure 6**). The OL/DA statistics in the figure are calculated using domain-averaged time series of OL/DA SWE over the effective surrounding areas by localization distances with the corresponding UA SWE. For the whole snow season that includes both accumulation and melting periods, the boxplot of the 1:1 slope shows that the localized DA SWE were improved as compared to OL. The slopes of the DA SWE are closer to 1 than the OL's slopes. The bias and RMSD boxplots also show that the DA SWE has lower errors than the OL SWE for all localization distances, expect for bias at 48 km which is too low

(median: – 23 mm). The OL's RMSDs slightly increased at the distances up to 16 km (median: 72 mm) and decreased after that, while the DA's RMSD values continually decreased with increasing the distances up to 48 km (median: 53 mm). When the statistics were calculated for the accumulation and melting periods separately, the lower RMSDs and slopes closer to 1 of the localized DA SWE were found consistently. As previously discussed, the efficacy of assimilating the airborne gamma SWE is greater during the accumulation period, especially for bias and RMSD, than during the melting period. In the melting

period, the improvements in the RMSD and 1:1 slope with longer distances are achieved, even though biases were consistently negative due to early melting.

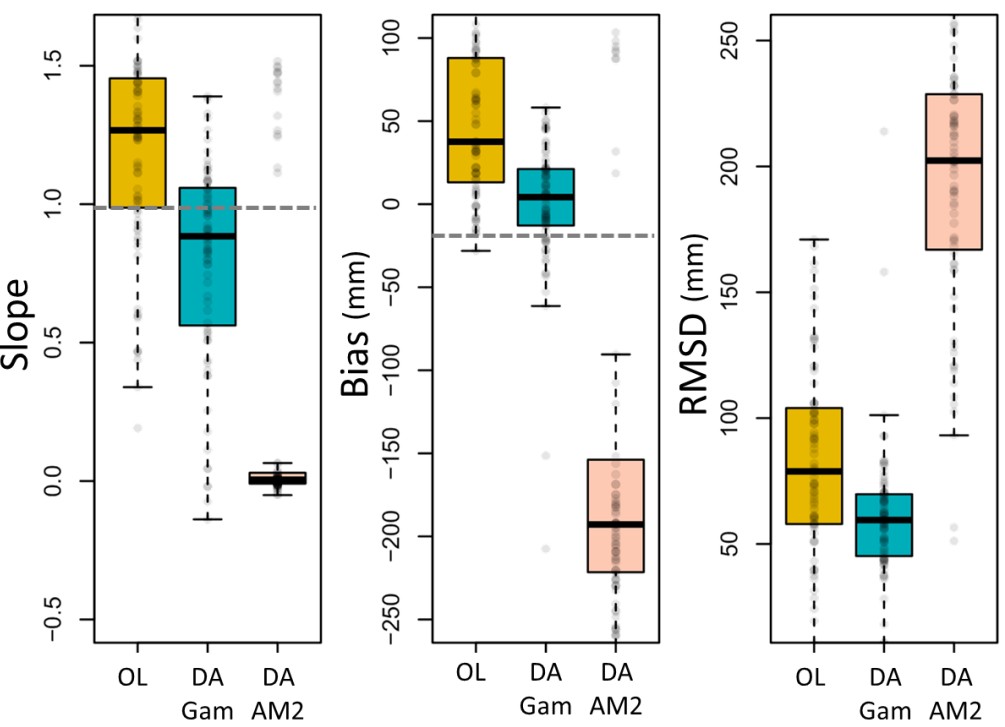

**Figure 7**. Comparison of the SWE estimation performance between the open-loop (OL), gamma DA, and AMSR2 DA as compared to the UA SWE at the 16-km localization distance for the mutual DA effective accumulation periods.

To compare the AMSR2 DA outputs to localized gamma DA outputs, we chose the gamma DA outputs at the 16-km localization distance which have similar effective spatial coverages of DA around gamma flight lines (10 km AMSR2

resolution and 16-km localization; **Figure 7**). Because the AMSR2 SWE was largely underestimated in the study domain (see **Figure 4**), assimilating the AMSR2 SWE measurements did not improve the modeled SWE estimates. All error metrics of the AMSR2 DA SWE were degraded (e.g., median bias: -193 mm and RMSD: 202 mm) as compared to the OL (median bias: 38 mm and RMSD: 79 mm). The localized gamma DA SWE performance is clearly improved based on the error metrics (median bias: 4 mm and RMSD: 59 mm). The positive biases and high slopes of the OL SWE were improved, and the RMSD also decreased approximately by 20 mm.

Overall, we found that the localized DA using the airborne gamma SWE observations improved the model SWE up to 32 km distances, which is supported by the recent study that a single gamma SWE observation spatially represents up to 50 km even in dense temperature forest environments (Cho et al., 2022). The study found that there was strong agreement between the gamma SWE observations and in-situ snow course transects (R-value: 0.78; RMSD: 53 mm) at distances up to 50 km in the northeastern U.S. The results in this study indicate that, even though the airborne gamma SWE measurements exist with limited spatial coverages, the combined use of the physical model and DA with the gamma SWE has a potential to improve regional estimations of the SWE.

## 6. Discussion

We observed two issues associated with the Noah-MP SWE estimates in the study domain: 1) Noah-MP considerably overestimated SWE during the snow accumulation period; while 2) it underestimated SWE (i.e., early snowmelt) during the snow ablation period (see **Figure 4**). The former issue was mitigated through the assimilation of the gamma SWE retrievals, whereas the latter issue was not. These issues can be attributed to parameterization schemes and/or atmospheric forcing employed in Noah-MP. Parameterization options for the precipitation phase partitioning method, ground surface albedo, surface layer drag coefficient, and snow/soil temperature time scheme can affect the snow simulations (You et al., 2020).

To further analyse the issues, we conducted additional experiments using different parameterization schemes and atmospheric forcing. That is, the BATS scheme for partitioning precipitation into rainfall and snowfall, CLASS scheme for snow albedo, Chen97 scheme for surface layer drag coefficient, fully-implicit snow and soil temperature time scheme, and the bias-corrected MERRA-2 forcing were additionally tested. As shown in **Figure 8**, the use of BATS or CLASS snow albedo schemes do not make a significant difference in the SWE estimates as compared to the original OL results. Although the Chen97 or fully-implicit schemes are effective in delaying the snow removal date, they add considerably more snow during the snow accumulation period and do not help capture snowmelt start date (**Figure 8**). Furthermore, the effectiveness of each parameterization scheme varies with flight lines and time periods within the study domain as also emphasized by You et al. (2020). Figure 8 shows that the use of the bias-corrected MERRA-2 forcing is effective in improving the SWE estimates during the snow accumulation period, but did not improved the issue of rapid snow melting. The combined use of the bias-corrected MERRA-2 forcing and the fully-implicit scheme leads to improved snow removal timing, but largely overestimated SWE

during the snow accumulation period. We originally used the uncorrected MERRA-2 forcing to demonstrate the feasibility of the gamma SWE DA for improving the model estimates of SWE, particularly in forested areas, using the atmospheric forcing as is (i.e., without bias-correction), which is a typical case of operational prediction or monitoring systems. Here, it is worth noting that assimilation of the gamma SWE data provides similar SWE estimates to the case of using the bias-corrected forcing

with a semi-implicit scheme when the gamma SWE observations are available during the snow accumulation period.

# a. OL with different parameterization schemes

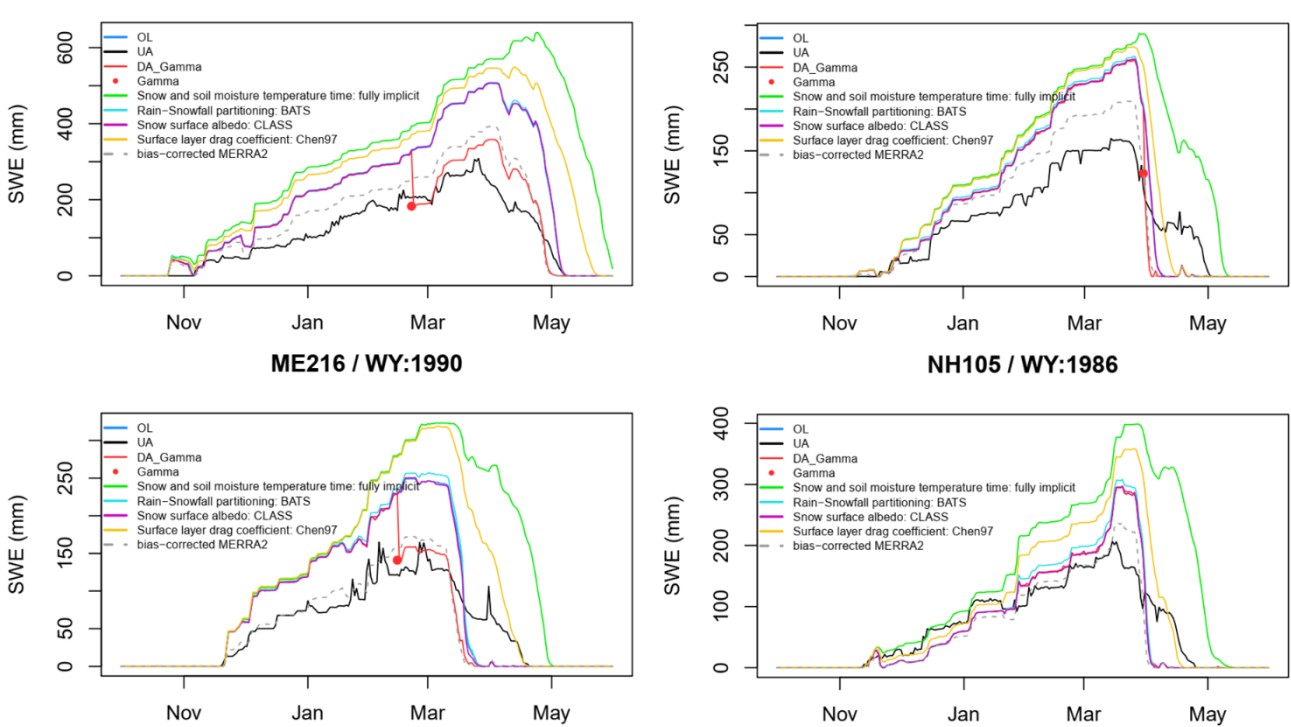

# b. DA + bias-corrected MERRA2 with semi vs. fully implicit

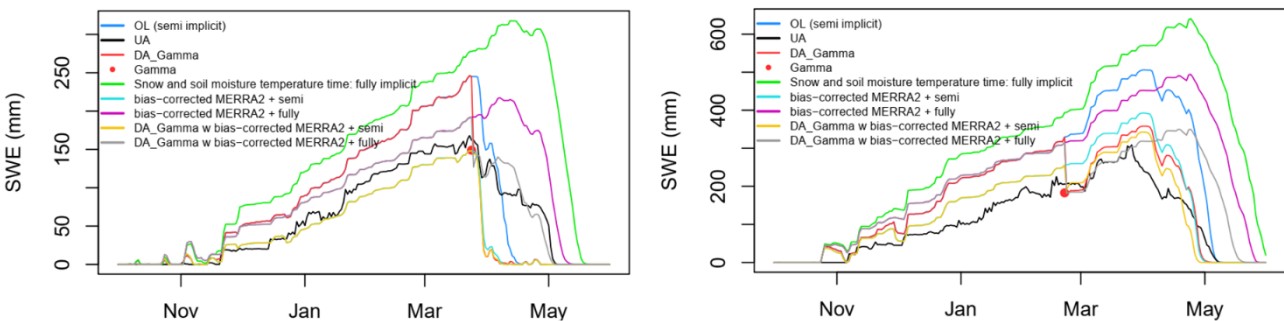

**Figure 8**. Examples of SWE time series including the additional open-loop (OL) experiments using different parameterization schemes related to snow simulations such as the snow/soil temperature time scheme (semi-implicit vs. fully-implicit), partitioning precipitation phase (Jordan91 vs. BATS), ground surface albedo (BATS vs. CLASS), and surface layer drag coefficient (Monin-Obukhov [M-O] similarity theory vs. Chen97 [original Noah]), and (b) DA runs forced by original vs. bias-corrected MERRA2 forcings with each snow/soil temperature time scheme which is a parameterization option largely affecting snow simulations.

Many studies (e.g., Aoki et al., 2011; Augas et al., 2020; Cheng et al., 2008; Jennings et al., 2018; Kwon et al., 2014; Lecomte et al., 2011; Livneh et al., 2010; Saha et al., 2017; Suzuki and Zupanski, 2018) have emphasized the importance of the number of snow layers for accurate estimates of snowmelt timing because of its impact on the vertical snow temperature gradient. Augas et al. (2020) demonstrated that the accuracy of the SWE estimates increases with more snow layers, and Lecomte et al. (2011) showed that the agreement between the observed and modeled vertical snow temperature gradient is improved by adding more snow layers. The minimum threshold of the number of snow layers suggested by existing studies ranges from 3 to 20 depending on locations, periods, and model setup. Different precipitation partitioning methods may lead to differences in the amount of snowfall and subsequent snowpack (Jennings et al., 2018; Letcher et al., 2021; Suzuki and Zupanski, 2018; Xia et al., 2017), even though there were no significant differences in SWE between the two schemes. We used the scheme of Jordan (1991), in which total precipitation is fractionally divided into rainfall and snowfall using two thresholds of air temperature (i.e., no snowfall when Tair > 2.5°C; all precipitation is snow when $T_{air} \leq 0$°C; and fractional snowfall when 0°C < $T_{air} \leq 2.5$°C). However, Noah-MP uses a spatially uniform threshold of $T_{air}$. Jennings et al. (2018) found that rain-snow $T_{air}$ thresholds exhibited significant spatial variability across the Northern Hemisphere with the warmest thresholds in continental and mountain areas while with the coolest thresholds in maritime areas and lowlands. This implies that the high $T_{air}$ threshold (i.e., 2.5°C) used in Noah-MP may lead to the overestimated snowfall, and subsequently the overestimated snow depth and SWE as the study area is characterized by maritime. Letcher et al. (2021) demonstrated that the use of cooler $T_{air}$ thresholds in Noah-MP can improve the estimates of peak SWE in the northeastern United States. To verify this, four Noah-MP SWE simulations with Jordan (1991)'s scheme and a single threshold of 0°C with two different meteorological forcings (MERRA2 and the North American Land Data Assimilation System; NLDAS2) are compared to ground-based SWE observations from Oct 1, 2002, to May 31, 2003, at Hubbard Brook, New Hampshire, which is within the study domain (Figure 9). This supports the previous finding that the overestimated SWE with Jordan's scheme was reduced with a single threshold of 0°C for both forcings. This also presents that the use of regionally reliable meteorological forcings (e.g., precipitation) generates accurate SWE estimations. At the same time, further improvement in the modeled SWE during the melting season can be achieved by employing more sophisticated snow models since the sophisticated snow models with multi-layer of snowpack take into account meltwater infiltration and refreezing within the snowpack (Avanzi et al., 2016; Terzago et al., 2020).

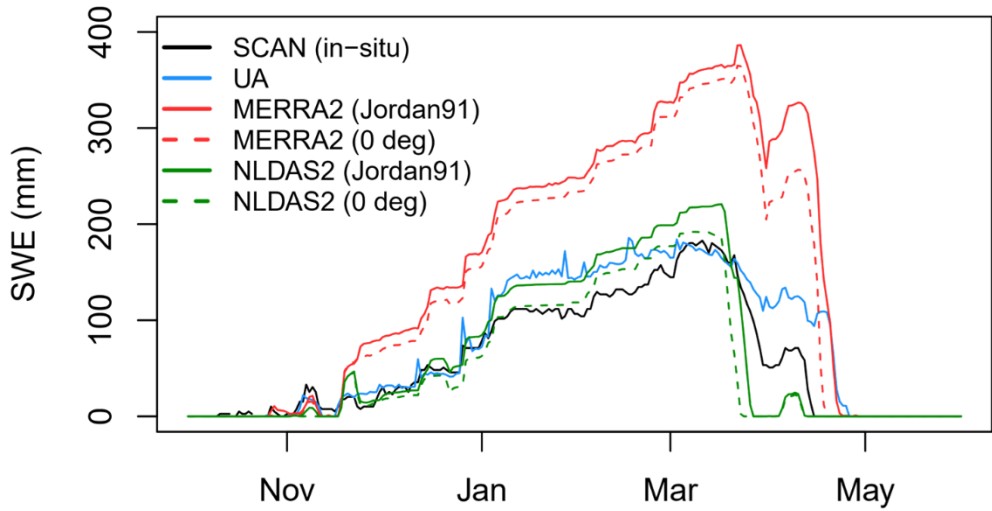

**Figure 9**. Comparison of SWE time series between four Noah-MP simulations and the Soil Climate Analysis Network (SCAN) ground-based observations at Hubbard Brook, New Hampshire from Oct 1, 2002, to May 31, 2003 (https://wcc.sc.egov.usda.gov/nwcc/site?sitenum=2069). The four Noah-MP SWE simulations were generated with Jordan (1991)'s scheme and a single threshold of 0°C and two meteorological forcings (MERRA2 – which is used for OL and the North American Land Data Assimilation System; NLDAS2), respectively.

While the model parameter calibration is not conducted here because it is outside the scope of the current study, we acknowledge that the parameter calibration procedure could further improve the model performance for regional applications. Cuntz et al. (2016) provided a sensitivity analysis of the Noah-MP parameters including both the adjustable and hard-coded parameters that affect simulations of hydrological processes. Based on their analysis, some snow-related Noah-MP

hydrological simulations exhibit high sensitivity to hard-coded parameters rather than tunable parameters. For example, snowmelt-induced surface runoff is sensitive to hard-coded snow-related parameters for surface resistance, partitioning of incoming radiation into direct and diffuse radiation, and snow thermal conductivity. Meanwhile, the current DA framework does not perform assimilation if one or more of the prior model ensemble members do not have snow. Thus, the gamma SWE retrievals could not add value to the SWE estimates during the snow melting period. To address this issue, a rule-based

approach (e.g., Kwon et al., 2019), that adds a thin snow layer when the model simulates snow-free conditions, but observations have snow, can be explored in a future study.

It is possible that the inherent uncertainties in the gamma radiation method limit the potential improvements through DA even though the airborne gamma radiation SWE was beneficial to enhance SWE estimations by assimilating into Noah-MP land surface models. The potential sources of error in the gamma SWE retrievals have been explored in previous findings

(Carroll and Carroll, 1989a, 1989b; Glynn et al., 1988; Offenbacher and Colbeck, 1991). An impact of forest biomass on the accuracy of airborne gamma SWE measurements has been examined over forested watersheds (Carroll and Vose, 1984; Vogel et al., 1985). Carroll and Vose (1984) presented that there was 23 mm of RMSE between airborne gamma SWE and in-situ SWE for the moderate snowpack (20 to 470 mm of in-situ SWE) in Lake Superior and Saint John basins, New Brunswick,

Canada. Spatial variability in elevations over the gamma flight footprint can cause larger errors in SWE (Cho et al., 2020b; Carroll and Carroll, 1989b; Cork and Loijens, 1980). Cho et al. (2020b) found that heterogeneous characteristics (e.g. elevation range and slope) within a flight line cause underestimates of gamma SWE as compared to UA SWE. Cork and Loijens (1980) discussed that the measurements of the attenuation of the gamma count rate over the snowpack with its large spatial variability were systematically underestimated leading to the SWE underestimation. Because the results use the NOAA standard gamma radiation SWE retrievals without manual corrections, the DA results would be improved with the updated gamma SWE products in regions by correcting the existing potential errors. Lastly, the spatiotemporal sparseness of the airborne gamma SWE observations due to the operational costs is an inherent issue that may limit the widespread use of gamma SWE observations for DA work. However, as supported by our findings, effective uses of the gamma SWE (e.g. localization function) can enhance the utility of the gamma SWE into the DA framework.

**7 Conclusion and Future Perspectives**

In the snow hydrology community, DA has been used as a promising approach to improve SWE estimation at a large spatial scale by merging remote sensing observations with LSM predictions. In densely forested regions, however, most remote sensing techniques have limited performance of SWE due to attenuating or/and scattering radiation signals by canopy (e.g. passive microwave $T_B$ and Lidar), resulting in large uncertainty in DA outputs. The historically well-established, airborne gamma radiation technique has provided a strong potential in wet snow and dense forest conditions because the gamma approach uses an attenuation difference in the terrestrial gamma-ray emission by water in the snowpack (any phase) between snow-off and snow-on conditions. In this study, the airborne gamma SWE observations are assimilated with the Noah-MP model's SWE in densely forested regions in the northeastern U.S. We found that the assimilation of the airborne gamma SWE observations enhanced the model SWE estimates despite the limited number of the measurements (up to four SWE values during a winter period). The added value of the gamma data on the model SWE estimates was greater for the relatively lower VCF range. While the gamma-based DA SWE had relatively lower improvement in areas with higher topographic heterogeneity, the DA SWE with reduced errors was found as compared to the OL. We also found that the localized DA with the gamma SWE observations with distances up to 32 km reduced the model SWE's errors, indicating the gamma SWE has a potential to improve regional estimations of the SWE and subsequently snowmelt runoff. Despite the accuracy of the gamma data on the DA framework, the improvements were limited by the spatial and temporal sparseness of the gamma measurements. With the enhanced physics in LSMs and optimal uses of the gamma data using enhanced DA/interpolation methods, future studies may achieve a further improvement of the modeled SWE for larger areas where gamma flights do not exist.

*Data availability*. The original airborne gamma radiation SWE data are available from the NOAA NWS NOHRSC website (http://www.nohrsc.noaa.gov/snowsurvey/). The reformatted airborne gamma SWE data (NetCDF format), the R codes used to reformat them, and SWE outputs from the OL and DA runs are available at http://www.hydroshare.org/resource/fc5c757899fb49a5869e597451120a33 (Cho et al., 2023). The UA daily 4-km SWE data

(Version 1) and JAXA AMSR2 L3 Global Daily 10 km SWE data (Version 1) are available from the website (https://nsidc.org/data/nsidc-0719 and https://gportal.jaxa.jp/gpr/information/download, respectively). The MERRA2 forcing dataset is distributed by the NASA Goddard Global Modeling and Assimilation Office (GMAO; https://gmao.gsfc.nasa.gov/reanalysis/MERRA-2/data_access/). To replicate the land surface model simulation and data assimilation, users can use the NASA Land Information System which is freely available at https://github.com/NASA-LIS/LISF. The lis-config files used in this study are available at the above repository.

*Author contributions*. EC and YK conceptualized the research, did the formal analysis, and wrote the initial draft. SVK and CMV helped with the investigation, provided technical and scientific inputs, supervised the project, and reviewed and edited the paper.

*Competing interests*. On behalf of all authors, the corresponding author states that there is no conflict of interest.

*Acknowledgments*. The authors gratefully acknowledge support from NASA Terrestrial Hydrology (THP) Program (NNH16ZDA001N). We are grateful to the NOAA NWS NOHRSC colleagues (Tom Carroll, Don Cline, and Carrie Olheiser) for the dedication and steady efforts to operate the airborne gamma snow survey. Computing resources to run the NASA land information system (LIS) were supported by the NASA Center for Climate Simulation.

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
