# Peer review of "Assimilation of airborne gamma observations provides utility for snow estimation in forested environments"

_Hydrology and Earth System Sciences, 2022_

## Referee Comment (RC1)

**Review of Assimilation of airborne gamma observations provides utility for snow estimation in forested environments**

**Recommendation Major Revisions:**

The authors have put forth an interesting study that aims to quantify the impact of assimilating airborne gamma SWE measurements into the Noah-MP land surface model.  This study focuses on forested regions in the Northeast United States, a region that is underrepresented in snow-hydrology research.  Furthermore, the choice to assimilate airborne SWE measured from gamma radiation into Noah-MP and quantify the impacts is somewhat novel.  The use the LIS, airborne gamma measurements and UA datasets is appropriate, and the paper is generally well structured.  Finally, this topic of study is interesting and the results are of clear value to both the research community and regional stakeholders.  Accordingly, this study fits well within the journal scope and is worthy of publication.  However, I see two major deficiencies with in the work that need to be addressed prior to publication.

First, I do not see how any reasonable person would be able to replicate this work based on the information provided in the methods section and the data availability statement.  I think this needs to be addressed, especially considering that many journals are moving towards an increased emphasis on replicability and open data practices.  In my view, significant effort is required to modify the method section, and more data should be made available (if not large model output datasets, at least model config files and namelists) before this study could be considered open and transparent.

The second major deficiency is that the open-loop (OL) Noah-MP simulation performs so poorly that I'm left wondering if the model was configured or forced properly.  This feeling is exacerbated by the fact that the authors provide very little information regarding the model configuration.  Further, the authors make several vague and dismissive statements pinning the poor performance of the model on "model physics" with no supporting evidence.  Considering that there are now dozens of studies that show that Noah-MP has rather good performance with respect to snow (including over the Northeast), the exceedingly poor model performance in this study is an outlier, and should be addressed.  I suspect that the poor model performance is related to the model configuration chosen for this specific study rather than issues intrinsic to the model.  So, prior to publication, the authors should revisit their baseline OL model configuration and track down some of the causes for the poor model performance.  My suspicion is that in doing so, they will be able to attain much better accuracy within the OL simulation, and accordingly, better and more robust results regarding the impact of the SWE DA. Once these two issues are addressed, I think the study will easily meet the criteria for publication and make a wonderful additional to the snow-hydrology literature.

**Major Specific Comments:**

**Line 175 – 190:**  This section could be improved with additional information regarding the Noah-MP configuration.  There are several specific questions that I have regarding the model set up that may go towards explaining some of the results in the results section, particularly with regards to the melt season:

1) Was the TCF described in section 3.4 used to inform the Noah-MP vegetation fraction? And if not, was the Noah-MP vegetation fraction used in the model compared to the TCF for consistency?
2) Was forcing data downscaled to the terrain at all?
3) Similarly, was Noah-MP run on a grid covering a region? Or ONLY for "grid-cells" that would have assimilated data, i.e., along the flight-path + gridcells located within the localization radius?
4) What were the physics parameterizations? It's mentioned later on in the paper that the Jordan phase partitioning is used, but equally as relevant would be the snow albedo option (BATS vs. CLASS), the melt-factor used in determining the pixel snow-cover-fraction, the radiation_scheme option (specifically whether or not Fveg is used) and the temp_time_scheme option. (Cuntz et al, 2016 and You et al. 2022 illustrate the impact of some of these options on snow).
5) What went into the ensemble members? Presumably, this was an ensemble with perturbed initial / forcing conditions? Or was it a physics ensemble?

Towards this end, the data-availability statement of "To replicate the land surface model simulation and data assimilation, users can freely access …" is **entirely insufficient** for replicating the model simulation. At a *bare minimum*, any/all lis.config files used to run the simulations should be included somewhere as supplementary material. Further, was this in off-the-shelf version of LIS? Or one that was specifically modified to assimilate airborne gamma snow observations?

**Lines 227 – 229:** "*Figure 2 was a consequence of the fact that the overestimated SWE during the accumulation season and early in the melt season was offset by the underestimated SWE during the snowmelt season (i.e., April and May)*"

The statement appears generally true looking at the time-series data, but that raises more pressing questions about the OL config. To me it looks like the OL model config has serious issues that cancel each other out in the bulk sense and leads a reasonable overall bias, were an assimilation of SWE can correct one of those issues, leading to worse results. Regardless, the fact that the OL simulation often shows a peak SWE over-estimate greater than 100mm (50-80% of observed) *and still* melts out 2-4 weeks before the observed melt-out date is a red flag. In particular, the time-series for WY 1989 and WY 1997 are concerning. I *strongly recommend* that the authors a) contextualize the poor performance of the OL simulation with other recent studies evaluating Noah-MP snow in this region (e.g., Letcher et al. 2022 or Sthapit et al. 2022), and/or b) attempt to track down and correct the source of the poor performance. My experience is that the Noah-MP performance is strongly tied to the quality of the forcing data, so I recommend at least doing some analysis comparing the MERRA forcing to in-situ data in the region. Really, any efforts to try and understand why the OL model is performing so poorly would improve the manuscript quality. My feeling is that, if Noah-MP is configured well and driven with good forcing data, the OL performance would be almost certainly be much better than this paper suggests. If the OL simulation is configured for good performance, the impact

of the DA will be more accurate and robust, so it is well worth the effort to try and improve the baseline simulation.

**Line 243 – 248:** This whole section needs additional analysis. The conclusions presented by the authors seem to make no effort in gathering evidence to support them.  Rather, speculative comments like "probably due to limited model physics" or "may be attributed to model structure physics" are used to explain why the DA didn't help much during the spring.  I don't think these types of statements really belong in a results section, especially when there are ample analyses that could have been done to support or refute them.  Considering that there are several studies showing fairly decent Noah-MP performance, I don't think the poor performance found here is attributable to intrinsic issues with the model architecture.

**"**However, the assimilation of the airborne gamma SWE measurements was not able to improve the snow ablation timing probably due to temporally sparse gamma data as well as limited model physics"

I disagree with this statement; I and I see very little evidence in the paper supporting it.  For instance, WY2015 seems to have a gamma-observation during the ablation season, and this does not improve the simulation at all.

**General comment:** Have the authors considered other metrics for quantifying snow improvement such as peak SWE amount and timing? Or melt out date?  RMSE can be a tricky metric to quantify snow, since snow has a lower bound of zero (i.e., a lot of data points will simply be comparing zeros to each-other). See Rhodes et al. (2018) and Trujillo and Molotch (2014).

**References:**

Cuntz, M., Mai, J., Samaniego, L., Clark, M., Wulfmeyer, V., Branch, O., Attinger, S. and Thober, S., 2016. The impact of standard and hard-coded parameters on the hydrologic fluxes in the Noah-MP land surface model. Journal of Geophysical Research: Atmospheres, 121(18), pp.10-676.

Rhoades, A.M., Jones, A.D. and Ullrich, P.A., 2018. Assessing mountains as natural reservoirs with a multimetric framework. Earth's Future, 6(9), pp.1221-1241.

Sthapit, E., Lakhankar, T., Hughes, M., Khanbilvardi, R., Cifelli, R., Mahoney, K., Currier, W.R., Viterbo, F. and Rafieeinasab, A., 2022. Evaluation of Snow and Streamflows Using Noah-MP and WRF-Hydro Models in Aroostook River Basin, Maine. Water, 14(14), p.2145.

Letcher, T.W., Minder, J.R. and Naple, P., 2022. Understanding and improving snow processes in Noah-MP over the Northeast United States via the New York State Mesonet.

Trujillo, E. and Molotch, N.P., 2014. Snowpack regimes of the western United States. Water Resources Research, 50(7), pp.5611-5623.

You, Y., Huang, C., Yang, Z., Zhang, Y., Bai, Y. and Gu, J., 2020. Assessing Noah-MP parameterization sensitivity and uncertainty interval across snow climates. Journal of Geophysical Research: Atmospheres, 125(4), p.e2019JD030417.

**Minor comments:**

There are numerous instances of RMSD and a couple of instances of RMSE throughout the paper, so my assumption is that RMSD is preferred. I'm assuming that RMSD is Root Mean Square Difference? Either one is fine, but please be consistent, and consider spelling out RMSD in the paper when it's first used.

**Line 206:** Was the model snow LWC content updated proportionally? Or was any SWE added/subtracted through assimilation considered all snow? E.g., if the snowpack was 5% liquid, was SWE added -> LWC= LWC + 0.05*NEW and FROZ = FROZ + 0.95*NEW?

**Line 225 -226:** Did the authors mean: "closer to" instead of "closed to?" Since the authors include the median slope of the linear regression for the OL, would it make sense to also include it from the DA run?

**Lines 226-227:**
 Was Figure 2 only for grid-cells that received DA updates? OR ALL gridcells (see previous comment on model config)?

**Lines 226-227:** I recommend rewording: "The lower bias of the SWE estimates from the OL as compared to the DA in Figure 2 …" to be more specific: "The absolute SWE bias was higher in the DA simulation compared to the OL simulation (Figure 2b). However, this was a consequence of the fact …."

**Line 237:** Please replace the word "enhanced" with "improved"

**Line 240:** Recommend deleting "As shown in the figure"

**Section 5.2 throughout:**
I recommend replacing references to the "lower" and "higher" groups with "low" and "high" since this is how the groups were introduced.

**Line 252:** What is "differences in the 1:1 slope?" is it differences in the linear-regression slope between the OL and DA simulations? Or is it differences between the linear-regression slope AND the 1-to-1 line for each simulation? Please clarify.

**Figure 4:** I suggest increasing the font size on the legend to be more readable.

**Line 253 – 255:** Consider rewording: "In the figure, two groups of each land surface characteristics were determined by dividing the gamma flight lines into two (i.e.,low and high) groups of equal numbers of the flight lines."

As:

"In this analysis, the land-characteristics sampled by the gamma flights were divided equally into two groups (i.e., low and high) such that the land characteristic value separating the two groups allowed for equal numbers of samples in each group."

**Line 260**: what does "updated" mean here?  Improved?  Changed?  Please clarify.

**Line 264:** "added value of the gamma SWE data on the model SWE estimates via assimilation" This is really awkward wording, please rephrase.

**Figure 6:** I'm a little confused how the mean slope for the 4km localized OL simulation is near 1-to-1 as compared to 1.8/1.9 shown in figure 3.  I'm also confused as to why there is an increase in the error metrics in the OL simulation as the localization distance is increased?  Does that imply that the model is better over the flight-lines, even without any DA?  Am I just interpreting this analysis wrong? Is it just that the 4km localization provides fewer grid points, and therefore fewer opportunities for the model to accumulate large errors? Also, if the OL slope for 4km is close to one, how does that correspond to such high (50-60mm) bias?  I don't think the y-intercept would explain that.

My understanding is that the model is running Noah-MP along some flight-path, and some number of adjacent gridcells corresponding to various different radii of influence.  Shouldn't the OL metrics be independent of how many gridcells are in the simulation?  Either way, the methods here are pretty vague, I recommend more clearly explaining the experiment to eliminate confusion.

**Line 280:** "closed" should be "close"

**Line 307-308:** "The overestimated SWE during the snow accumulation period was likely attributed to a precipitation phase partitioning method employed in Noah-MP."

This could very well be true, but this is something that could be easily tested by running a couple of simulations with different temperature thresholding (e.g., PCP_PARTITION_OPTION = 3, instead of 1 in the namelist).  That way the authors wouldn't have to speculate here.

**Lines 322 – 324:** "Although the snowpack in Noah-MP can have up to three snow layers, it may not be enough to accurately reproduce the energy budget within the snowpack in the study area."

Do you have a citation to support this statement? While there is probably at least some truth to it, I'm not sure that I agree 100%. In terms of simulating bulk SWE, I don't often see significant improvement with the addition of *more* snow-layers, just so long as the model has multiple discrete snow layers in it.

**Line 344 – 345: "**However, as proven in our findings, effective uses of the gamma SWE (e.g. localization function) will maximize the utility of the gamma SWE into the DA framework."

I would scale back the certainty in this statement: e.g., replace "proven in" with "supported by" and "will maximize" with "can enhance."

**Lines 347- 348:** "promising alternative"; Alternative to what?

**Lines 356 – 358:** "The added value of the gamma data on the model SWE estimates was greater for the relatively lower VCF range. For areas with higher topographic heterogeneity, the gamma-based DA SWE was still effective in reducing the errors."

This is very awkwardly worded, the first sentence is in reference to the vegetation fraction, and the second was for the topography heterogeneity. Specifically, the use of the phrase "still effective" is confusing since there is no "base-state" effectiveness of the DA for terrain heterogeneity described here in the conclusions. Please reword.

**Lines 361 – 362:** "uncertainties in the Noah-MP physics (i.g. precipitation partitioning and simplified snow layers)."

I really do not think that this statement belongs in the conclusions, since it's just speculation on the part of the authors.

**Data Availability:**
Definitely a lot to be desired here, by only pointing readers to publicly available datasets, and choosing not to include model configuration data, analysis code, or processed (example) data, there is simply no way that a reasonable person would be able to repeat the experiment and analysis described in the paper. At a bare-minimum, the model configuration files should be included, as should any code that processes the data prior to being fed into the model.

**Author Contributions:**
I'll leave this to the editors, but I'm not sure "provided the funding" is a good reason to include for co-authorship. While in this instance it doesn't matter since SVK and CMV provided other input that would merit co-authorship, explicitly listing "provided funding" seems like it could be problematic.

---

## Author Comment (AC1)

**Reviewer 1**
**Recommendation Major Revisions:**
The authors have put forth an interesting study that aims to quantify the impact of assimilating airborne gamma SWE measurements into the Noah-MP land surface model. This study focuses on forested regions in the Northeast United States, a region that is underrepresented in snow-hydrology research. Furthermore, the choice to assimilate airborne SWE measured from gamma radiation into Noah-MP and quantify the impacts is somewhat novel. The use the LIS, airborne gamma measurements and UA datasets is appropriate, and the paper is generally well structured. Finally, this topic of study is interesting, and the results are of clear value to both the research community and regional stakeholders. Accordingly, this study fits well within the journal scope and is worthy of publication. However, I see two major deficiencies within the work that need to be addressed prior to publication.
[Answer] Thank you for your constructive feedback and the valuable comments on our manuscript. We have carefully revised our manuscript based on each of your comments.

First, I do not see how any reasonable person would be able to replicate this work based on the information provided in the methods section and the data availability statement. I think this needs to be addressed, especially considering that many journals are moving towards an increased emphasis on replicability and open data practices. In my view, significant effort is required to modify the method section, and more data should be made available (if not large model output datasets, at least model config files and namelists) before this study could be considered open and transparent.
[Answer] We appreciate the reviewer for pointing the limited data-availability statement. Based on the Reviewer's comment, we will provide all lis.config files used to run the simulations in supplemental materials. And original and reformatted airborne gamma SWE data (as well as the R code used to reformat them) as well as time series outputs from the OL and DA runs will be available for download at [will add a link to data from Zenodo, currently being setup with an ODC Attribution (ODC-BY) license for access without restrictions].

The second major deficiency is that the open-loop (OL) Noah-MP simulation performs so poorly that I'm left wondering if the model was configured or forced properly. This feeling is exacerbated by the fact that the authors provide very little information regarding the model configuration. Further, the authors make several vague and dismissive statements pinning the poor performance of the model on "model physics" with no supporting evidence. Considering that there are now dozens of studies that show that Noah-MP has rather good performance with respect to snow (including over the Northeast), the exceedingly poor model performance in this study is an outlier, and should be addressed. I suspect that the poor model performance is related to the model configuration chosen for this specific study rather than issues intrinsic to the model. So, prior to publication, the authors should revisit their baseline OL model configuration and track down some of the causes for the poor model performance. My suspicion is that in doing so, they will be able to attain much better accuracy within the OL simulation, and accordingly, better and more robust results regarding the impact of the SWE DA. Once these two issues are addressed, I think the study will easily meet the criteria for publication and make a wonderful additional to the snow-hydrology literature.
[Answer] We very much appreciate the reviewer's comment. To address the Reviewer's comment, we conducted additional experiments using multiple different parameterization schemes, which were

included in the revised manuscript. Please see our detailed answer to your comment for "Lines 227-229" below.

**Major Specific Comments:**
**Line 175 – 190:** This section could be improved with additional information regarding the Noah-MP configuration. There are several specific questions that I have regarding the model set up that may go towards explaining some of the results in the results section, particularly with regards to the melt season:

1) Was the TCF described in section 3.4 used to inform the Noah-MP vegetation fraction? And if not, was the Noah-MP vegetation fraction used in the model compared to the TCF for consistency?
Thank you for your comment. We cannot directly compare between the MEaSUREs TCF map used in this study and the vegetation fraction maps (NCEP greenness fraction), because the MEaSUREs data is annual-based historical VCF data while Noah-MP model in this study used NCEP monthly greenness fraction climatology. The main reason why we used MEaSUREs TCF maps in this work is that the maps provide "annual" vegetation fraction cover from 1982 to 2016, enabling us to obtain VCF values at the year when the historical gamma SWE was collected.

To address the Reviewer's question, the spatial maps between the MEaSUREs map in 2016 (left) and NCEP max greenness fraction map (right) are provided below. As an additional source, we also provided UMD Hansen forest map.

[Figure]

2) Was forcing data downscaled to the terrain at all?

Yes. The MERRA-2 atmospheric forcing was spatially and temporally interpolated using bilinear and linear interpolation algorithms, respectively, within LIS before being fed to the LSM.

3) Similarly, was Noah-MP run on a grid covering a region? Or ONLY for "grid-cells" that would have assimilated data, i.e., along the flight-path + gridcells located within the localization radius?

Noah-MP was run on the whole study domain (i.e., northeastern United States). However, the analysis (i.e., performance comparison between the OL and DA cases) was conducted only for grid cells within the flight lines and within the localization radius for the DA experiments without and with applying the localization approach, respectively.

4) What were the physics parameterizations? It's mentioned later on in the paper that the Jordan phase partitioning is used, but equally as relevant would be the snow albedo option (BATS vs. CLASS), the melt-factor used in determining the pixel snow-cover fraction, the radiation_scheme option (specifically

whether or not Fveg is used) and the temp_time_scheme option. (Cuntz et al, 2016 and You et al. 2022 illustrate the impact of some of these options on snow).

We appreciate the reviewer for pointing this out and providing great references. Physical parameterization scheme options of Noah-MP v3.6 used in the current study are listed below: (1) dynamic vegetation for the vegetation option; (2) Noah-type soil moisture factor for stomatal resistance (Chen and Dudhia, 2001); (3) Ball-Berry canopy stomatal resistance scheme (Ball et al., 1987); (4) TOPMODEL-based runoff scheme; (5) simple groundwater scheme (SIMGM; Niu et al., 2007); (6) general Monin-Obukhov similarity theory (M-O; Brutsaert, 1982) for surface layer drag coefficient; (7) NY06 scheme (Niu and Yang, 2006) for supercooled liquid water (or ice fraction) in frozen soil; (8) NY06 scheme (Niu and Yang, 2006) for frozen soil permeability; (9) modified two-stream radiation transfer scheme (Yang and Friedl, 2003, Niu and Yang, 2004); (10) Biosphere-Atmosphere Transfer Scheme (BATS) for ground surface albedo (Yang and Dickinson, 1996); (11) Jordan91 scheme (Jordan, 1991) for partitioning precipitation into rainfall and snowfall; (12) original Noah scheme for lower boundary condition of soil temperature; and (13) semi-implicit snow and soil temperature time scheme. We included these in the revised manuscript in Section 4.1. Regarding the effects of different parameterization schemes on the results are further discussed in the later part of this document.

*"Physical parameterization scheme options used in the current study are listed below: (1) dynamic vegetation for the vegetation option; (2) Noah-type soil moisture factor for stomatal resistance (Chen and Dudhia, 2001); (3) Ball-Berry canopy stomatal resistance scheme (Ball et al., 1987); (4) TOPMODEL-based runoff scheme; (5) simple groundwater scheme (SIMGM; Niu et al., 2007); (6) general Monin-Obukhov similarity theory (M-O; Brutsaert, 1982) for surface layer drag coefficient; (7) NY06 scheme (Niu and Yang, 2006) for supercooled liquid water (or ice fraction) in frozen soil; (8) NY06 scheme (Niu and Yang, 2006) for frozen soil permeability; (9) modified two-stream radiation transfer scheme (Yang and Friedl, 2003, Niu and Yang, 2004); (10) Biosphere-Atmosphere Transfer Scheme (BATS) for ground surface albedo (Yang and Dickinson, 1996); (11) Jordan91 scheme (Jordan, 1991) for partitioning precipitation into rainfall and snowfall; (12) original Noah scheme for lower boundary condition of soil temperature; and (13) semi-implicit snow and soil temperature time scheme."*

5) What went into the ensemble members? Presumably, this was an ensemble with perturbed initial / forcing conditions? Or was it a physics ensemble? Towards this end, the data-availability statement of "To replicate the land surface model simulation and data assimilation, users can freely access …" is entirely insufficient for replicating the model simulation. At a *bare minimum*, any/all lis.config files used to run the simulations should be included somewhere as supplementary material. Further, was this in off the-shelf version of LIS? Or one that was specifically modified to assimilate airborne gamma snow observations?

Model uncertainty was implicitly represented by the ensemble spread (ensemble size of 20 was used in this study), which was generated by perturbing atmospheric forcing fields and model prognostic state variables with the assumption of a Gaussian distribution. Perturbation parameters applied during the OL and DA runs are summarized in the following table, which was included in the revised manuscript.

**Table 1**. Perturbation parameters applied to model prognostic state variables and atmospheric forcing fields during the OL and DA runs.

| Variable | Perturbation Types | Std dev | AR(1) | Cross correlations | | |
|---|---|---|---|---|---|---|
| Model prognostic state variables | | | | *SWE* | *SD* | |
| *SWE* | M | 0.01 | 3 hr | – | 0.9 | |
| Snow depth (*SD*) | M | 0.02 | 3 hr | 0.9 | – | |
| Atmospheric forcing fields | | | | *SW* | *LW* | *P* |
| Shortwave radiation (*SW*) | M | 0.3 | 1 day | – | -0.5 | -0.8 |
| Longwave radiation (*LW*) | A | 50 W m$^{-2}$ | 1 day | -0.5 | – | 0.5 |
| Precipitation (*P*) | M | 0.5 | 1 day | -0.8 | 0.5 | – |

M: multiplicative; A: additive; AR(1): first-order autoregressive temporal correlation.

As we mentioned above, we will provide all lis.config files used to run the simulations as supplemental materials. And original and reformatted airborne gamma SWE data as well as time series outputs from the OL and DA runs are available used in this paper are available for download at [will add a link to data from Zenodo, currently being setup with an ODC Attribution (ODC-BY) license for access without restrictions].

We used the publicly released version of LIS. The file format of the gamma-SWE observations was converted to one that the original LIS code can read.

**Lines 227 – 229:** "*Figure 2 was a consequence of the fact that the overestimated SWE during the accumulation season and early in the melt season was offset by the underestimated SWE during the snowmelt season (i.e., April and May)*"

The statement appears generally true looking at the time-series data, but that raises more pressing questions about the OL config. To me it looks like the OL model config has serious issues that cancel each other out in the bulk sense and leads a reasonable overall bias, were an assimilation of SWE can correct one of those issues, leading to worse results. Regardless, the fact that the OL simulation often shows a peak SWE over-estimate greater than 100mm (50- 80% of observed) *and still* melts out 2-4 weeks before the observed melt-out date is a red flag. In particular, the time-series for WY 1989 and WY 1997 are concerning. I *strongly recommend* that the authors a) contextualize the poor performance of the OL simulation with other recent studies evaluating Noah-MP snow in this region (e.g., Letcher et al. 2022 or Sthapit et al. 2022), and/or b) attempt to track down and correct the source of the poor performance. My experience is that the Noah-MP performance is strongly tied to the quality of the forcing data, so I recommend at least doing some analysis comparing the MERRA forcing to in-situ data in the region. Really, any efforts to try and understand why the OL model is performing so poorly would improve the manuscript quality. My feeling is that, if Noah-MP is configured well and driven with good forcing data, the OL performance would be almost certainly be much better than this paper suggests. If the OL simulation is configured for good performance, the impact of the DA will be more accurate and robust, so it is well worth the effort to try and improve the baseline simulation.

As the reviewer commented, the performance of the Noah-MP open-loop (OL) run can be affected by model parameters, parameterization schemes, and atmospheric forcing. One of the references suggested

by the reviewer in the comment #4 (i.e., Cuntz et al., 2016) provide sensitivity analysis of the Noah-MP parameters including both the adjustable and hard-coded parameters that affect simulations of hydrological processes. Based on their analysis, some snow-related Noah-MP hydrological simulations exhibit high sensitivity to hard-coded parameters rather than tunable parameters. For example, snowmelt-induced surface runoff is sensitive to hard-coded snow-related parameters for surface resistance, partitioning of incoming radiation into direct and diffuse radiation, and snow thermal conductivity. Furthermore, while model parameter calibration is an important procedure for regional applications, parameter values empirically determined for certain areas are in general not applicable to other areas, and recalibration requires extensive datasets and efforts. As the parameter calibration is outside the scope of our current study, we decided to keep using the default parameter values suggested by the Noah-MP developer. We acknowledged this limitation in the revised manuscript. Consequently, here we only conducted additionally experiments using different parameterization schemes and bias-corrected MERRA-2 forcing.

*"While the model parameter calibration is not conducted here because it is outside the scope of the current study, we acknowledge that the parameter calibration procedure could further improve the model performance for regional applications. Cuntz et al. (2016) provided sensitivity analysis of the Noah-MP parameters including both the adjustable and hard-coded parameters that affect simulations of hydrological processes. Based on their analysis, some snow-related Noah-MP hydrological simulations exhibit high sensitivity to hard-coded parameters rather than tunable parameters. For example, snowmelt-induced surface runoff is sensitive to hard-coded snow-related parameters for surface resistance, partitioning of incoming radiation into direct and diffuse radiation, and snow thermal conductivity."*

Based on You et al. (2020)'s analysis, we tested four parameterization schemes that possibly affect the performance of the Noah-MP OL run. That is, the BATS scheme for partitioning precipitation into rainfall and snowfall, CLASS scheme for ground surface albedo, Chen97 scheme for surface layer drag coefficient, and fully-implicit snow and soil temperature time scheme were additionally tested. As shown in the figure below, the use of BATS or CLASS schemes did not make a significant difference in SWE estimates as compared to our original OL results. Although the Chen97 or fully-implicit schemes were effective in delaying the snow removal date, they added considerably more snow during the snow accumulation period and did not help capture snowmelt start date. Furthermore, the effectiveness of each parameterization scheme varied with flight lines and time periods within the study domain as also emphasized by You et al. (2020).

The figure also shows that the use of bias-corrected MERRA-2 forcing was effective in improving the SWE estimates during the snow accumulation period, but it still has the issue of rapid snow melting. The combinational use of the bias-corrected MERRA-2 forcing and fully-implicit scheme led to improved snow removal timing, but largely overestimated SWE during the accumulation period. Here, it is worth to note that assimilation of the gamma SWE data provides similar SWE estimates to the case of using the bias-corrected forcing with semi-implicit scheme when the gamma SWE observations are available during the snow accumulation period.

We also tested DA experiments using the bias-corrected forcing and a parameterization scheme "semi-vs. fully-implicit" of snow and soil moisture temperature time scheme which was effective in delaying the melt out date. The DA figure below shows examples that DA performances. In 1989, the DA output using

the bias-corrected forcing with fully-implicit (grapy line) captured better the peak SWE and melt out date than the original DA using original MERRA2 forcing with semi-implicit (red line). However, the new DA output overestimated SWE (peak value as well) and estimated later melt out date in 1991.

Based on these results, we tone down our previous discussion and included further analysis and related sentences in the revised manuscript. However, we decided not to reconduct all DA experiments using the bias-corrected forcing and different parameterization schemes due to the following reasons. First, the bias-corrected forcing is not readily available for operational use. Our fundamental purpose is to apply the gamma SWE DA framework in operational systems. Thus, this study focuses on demonstrating the efficacy of the gamma SWE DA in improving the SWE estimates of a model driven by any atmospheric forcing as is. Second, the effectiveness of using different parameterization schemes is not constant for different fight lines and time periods within the study domain.

However, we definitely agree with the reviewer's comment that the proper model configuration and accurate forcing may result in more robust conclusions. We acknowledge the limitation of our study and provided more comprehensive discussion on the limitation in the revised manuscript as below.

[revised manuscript text omitted]

**Line 243 – 248:** This whole section needs additional analysis. The conclusions presented by the authors seem to make no effort in gathering evidence to support them. Rather, speculative comments like "probably due to limited model physics" or "may be attributed to model structure physics" are used to explain why the DA didn't help much during the spring. I don't think these types of statements really

belong in a results section, especially when there are ample analyses that could have been done to support or refute them. Considering that there are several studies showing fairly decent Noah-MP performance, I don't think the poor performance found here is attributable to intrinsic issues with the model architecture. **"However, the assimilation of the airborne gamma SWE measurements was not able to improve the snow ablation timing probably due to temporally sparse gamma data as well as limited model physics"**
I disagree with this statement; I and I see very little evidence in the paper supporting it. For instance, WY2015 seems to have a gamma-observation during the ablation season, and this does not improve the simulation at all.

We agree with the reviewer's comment. As mentioned in the previous comment, we conducted additionally experiments using different parameterization schemes and bias-corrected MERRA-2 forcing. Based on the results, we revised the sentences as follows in the revised manuscript as follows:

*"As shown in Figure 4, compared to the UA SWE, Noah-MP simulated earlier snow melt-out despite the overestimated snow accumulation, which may be attributed to the Noah-MP model structure and physics (e.g., simplified representations of snow layers), parameterization schemes, and/or atmospheric forcing. Discussions on this are provided in section 6. Also, the peak SWE cannot be corrected if a single gamma SWE exists only after the accumulation period. The availability of more frequent gamma observations during both the accumulation and snowmelt melt seasons could lead to further improvements in estimating SWE in the ablation period while the Noah-MP model snow layer representation and forcings needs to be enhanced."*

**General comment:** Have the authors considered other metrics for quantifying snow improvement such as peak SWE amount and timing? Or melt out date? RMSE can be a tricky metric to quantify snow, since snow has a lower bound of zero (i.e., a lot of data points will simply be comparing zeros to each-other). See Rhodes et al. (2018) and Trujillo and Molotch (2014).

Thank you for the comment. We had considered the metrics such as the amount of peak SWE and timing and melt out date. However, we concluded that providing time series graph itself would be better to compare snow patterns between OL and DL and quantify snow improvement because unlike typical mountain snow regimes (based on the given references), ephemeral snow characteristics were frequently found in those regions, and those characteristics generated multiple peak SWE and melt on dates (an example time series for NH101 line is provided below). In addition to providing time series, we calculated RMSD, slope, and bias to quantify overall SWE performance as period-averaged quantifications (The time series' figures for all gamma lines will be provided in Supporting Information).

[Figure]

**Minor comments:**

There are numerous instances of RMSD and a couple of instances of RMSE throughout the paper, so my assumption is that RMSD is preferred. I'm assuming that RMSD is Root Mean Square Difference? Either one is fine, but please be consistent, and consider spelling out RMSD in the paper when it's first used.

We edited and consistently used RMSD throughout the manuscript.

**Line 206:** Was the model snow LWC content updated proportionally? Or was any SWE added/subtracted

through assimilation considered all snow? E.g., if the snowpack was 5% liquid, was SWE added -> LWC= LWC + 0.05*NEW and FROZ = FROZ + 0.95*NEW?

The amount of the SWE update from the assimilation of the gamma SWE retrievals was added only to ice content of the bottom snow layer. Then, snow layer variables such as thickness, snow ice and liquid water content, and SWE of each snow layer were adjusted using the same methods as used in the Noah-MP's snow layer compaction, combination, and subdivision procedures. We added the explanations in the revised manuscript as follows:

*"Note that assimilation of gamma SWE updates only modeled SWE and the amount of the SWE update is added to ice content of the bottom snow layer. Then, snow layer variables such as thickness, snow ice and liquid water content, and SWE of each snow layer are adjusted using the same methods as used in the Noah-MP's snow layer compaction, combination, and subdivision procedures."*

**Line 225 -226:** Did the authors mean: "closer to" instead of "closed to?" Since the authors include the median slope of the linear regression for the OL, would it make sense to also include it from the DA run?

Yes. We edited this sentence as below.
*"The values of 1:1 slope were closer to 1 (A median slope of OL and DL were 1.45 and 0.91, respectively) and RMSD values decreased, even though negative biases were found."*

**Lines 226-227:** Was Figure 2 only for grid-cells that received DA updates? OR ALL gridcells (see previous comment on model config)?
As we mentioned above, Noah-MP was run on the whole study domain (i.e., northeastern United States). However, the performance comparison between the OL and DA cases was conducted only for grid cells within the flight lines and within the localization radius for the DA experiments without and with applying the localization approach, respectively. Figure 2 was based on grid-cells that received DA updates.

**Lines 226-227:** I recommend rewording: "The lower bias of the SWE estimates from the OL as compared to the DA in Figure 2 …" to be more specific: "The absolute SWE bias was higher in the DA simulation compared to the OL simulation (Figure 2b). However, this was a consequence of the fact …."
Thank you for the recommendation. Based on the Reviewer's suggestion, we reworded the statement as below.
*"The absolute SWE bias was higher in the DA as compared to the OL simulation (**Figure 2**). However, this was a consequence of the fact that the overestimated SWE during the accumulation season and early in the melt season was offset by the underestimated SWE during the snowmelt season (i.e., April and May)."*

**Line 237:** Please replace the word "enhanced" with "improved"
Edited.

**Line 240:** Recommend deleting "As shown in the figure"
Deleted.

**Section 5.2 throughout:**
I recommend replacing references to the "lower" and "higher" groups with "low" and "high" since this is how the groups were introduced.
Thank you for the recommendation. We edited these.

**Line 252:** What is "differences in the 1:1 slope?" is it differences in the linear-regression slope between the OL and DA simulations? Or is it differences between the linear-regression slope AND the 1-to-1 line for each simulation? Please clarify.
This means differences in linear regression slope for each simulation (OL and DA) with UA SWE.

**Figure 4:** I suggest increasing the font size on the legend to be more readable.
Thank you for the suggestion. We increased the font size. Please see the attached.

[Figure]

**Line 253 – 255:** Consider rewording: "In the figure, two groups of each land surface characteristics were determined by dividing the gamma flight lines into two (i.e.,low and high) groups of equal numbers of the flight lines."

As: "In this analysis, the land-characteristics sampled by the gamma flights were divided equally into two groups (i.e., low and high) such that the land characteristic value separating the two groups allowed for equal numbers of samples in each group."

Thank you so much for the suggested words, which are much clear to convey the meaning. We edited as suggested.

**Line 260**: what does "updated" mean here? Improved? Changed? Please clarify.
Improved.

**Line 264:** "added value of the gamma SWE data on the model SWE estimates via assimilation"
This is really awkward wording, please rephrase.
We rephrased this as below.
*"For individual physical characteristics, the improvement of the model SWE estimates via the gamma SWE assimilation are greater for the low VCF range based on both bias and RMSD."*

**Figure 6:** I'm a little confused how the mean slope for the 4km localized OL simulation is near 1-to-1 as compared to 1.8/1.9 shown in figure 3. I'm also confused as to why there is an increase in the error metrics in the OL simulation as the localization distance is increased? Does that imply that the model is better over the flight-lines, even without any DA? Am I just interpreting this analysis wrong? Is it just that the 4km localization provides fewer grid points, and therefore fewer opportunities for the model to accumulate large errors? Also, if the OL slope for 4km is close to one, how does that correspond to such high (50-60mm) bias? I don't think the y-intercept would explain that.
My understanding is that the model is running Noah-MP along some flight-path, and some number of adjacent gridcells corresponding to various different radii of influence. Shouldn't the OL metrics be independent of how many gridcells are in the simulation? Either way, the methods here are pretty vague, I recommend more clearly explaining the experiment to eliminate confusion.
Thank you for the valuable comment. When double-checking our calculation for Figure 6, we found a script error when calculating the error metrics. The error metrics were miscalculated using a "static" domain-averaged OL SWE time series with DA SWE time series at different localization distances. They should have been calculated with a domain-averaged OL SWE time series at a "corresponding" localization distance. We reviewed throughout the script, corrected the part, and recalculated the metrics which are shown in the new figure below.

The reason why the OL statistics were changed with increasing the localization distances in Figure 6 is that the effective distance used to calculate the domain-averaged OL SWE was changed according to the localization distance. For example, at a localization distance of 4 km, the domain-averaged OL SWE is calculated using gridded SWE values within an effective surrounding area (4 km distance from the gamma flight lines). At 32 km, the domain-averaged OL SWE was calculated based on larger number of the gridded SWE values within surrounding areas up to 32-km distance from the gamma flight line.

To clearly explain the localization, we edited and added descriptions in Section 4.3.

*"To quantify if the spatially sparse gamma SWE observations can improve the SWE estimates in the surrounding areas, where the observations are not available, we apply a distance-based localization method into the assimilation procedure. The localization is applied in the assimilation by weighting*

*distances from the flight lines (up to a specified localization distance; r) using the Gaussian decay-based localization method as follows:*

$$W = exp\left\{\frac{-d^2}{2\cdot\left(\frac{r}{2}\right)^2}\right\} \tag{7}$$

*where d is the distance between the updated grid cells (i.e., flight lines) and grid cells without observations within a specified localization radius r. The degree of the SWE updates for a grid cell from the assimilation is calculated using the localization weight (W) which is calculated based on the distance (d) from the updated grid cells overlapped with the flight line."*

The result parts of the manuscript related to the revised figure 6 were also revised as below.

*"For the whole snow season that includes both accumulation and melting periods, the boxplot of the 1:1 slope shows that the localized DA SWE were improved as compared to OL. The slopes of the DA SWE are closer to 1 than the OL's slopes. The bias and RMSD boxplots also show that the DA SWE has lower errors than the OL SWE for all localization distances, expect for bias at 48 km which is too low (median: – 23 mm). The OL's RMSDs slightly increased at the distances up to 16 km (median: 72 mm) and decreased after that, while the DA's RMSD values continually decreased with increasing the distances up to 48 km (median: 53 mm). When the statistics were calculated for the accumulation and melting periods separately, the lower RMSDs and slopes closer to 1 of the localized DA SWE were found consistently. As previously discussed, the efficacy of assimilating the airborne gamma SWE is greater during the accumulation period, especially for bias and RMSD, than during the melting period. In the melting period, the improvements in the bias, RMSD and 1:1 slope are also achieved up to 32 km"*

[Figure]

**Figure 6**. Localized data assimilation (DA) and open-loop (OL) Noah-MP SWE performances as compared to the UA SWE with different localization distances (e.g., 4, 8, 16, 24, 32, and 48 km) for the whole (accumulation and melting periods), accumulation, and melting periods, respectively.

**Line 280:** "closed" should be "close"
Done.
**Line 307-308:** "The overestimated SWE during the snow accumulation period was likely attributed to a precipitation phase partitioning method employed in Noah-MP." This could very well be true, but this is something that could be easily tested by running a couple of simulations with different temperature thresholding (e.g., PCP_PARTITION_OPTION = 3, instead of 1 in the namelist). That way the authors wouldn't have to speculate here.

We additionally tested the Noah-MP OL run using different parameterization scheme (i.e., BATS) for partitioning precipitation into rainfall and snowfall. However, the use of the BATS scheme could not improve the SWE estimates during the accumulation period. Our additional experiments using the bias-corrected MERRA-2 forcing show that atmospheric forcing was the main source of the largely overestimated SWE during the snow accumulation period rather than the parameterization scheme. We revised the manuscript based on these additional analysis as already presented in the previous comments.

**Lines 322 – 324:** "Although the snowpack in Noah-MP can have up to three snow layers, it may not be enough to accurately reproduce the energy budget within the snowpack in the study area."
Do you have a citation to support this statement? While there is probably at least some truth to it, I'm not sure that I agree 100%. In terms of simulating bulk SWE, I don't often see significant improvement with the addition of *more* snow-layers, just so long as the model has multiple discrete snow layers in it.

We do not have references specific to Noah-MP applications, but there are many studies (e.g., Aoki et al., 2011; Augas et al., 2020; Cheng et al., 2008; Jennings et al., 2018; Lecomte et al., 2011; Livneh et al., 2010; Saha et al., 2017) that emphasize the importance of the number of snow layers for accurate estimates of snowmelt timing because of its impact on the vertical snow temperature gradient. Augas et al. (2020) demonstrated that the accuracy of the SWE estimates increases with more snow layers, and Lecomte et al. (2011) showed that the agreement between the observed and modelled vertical snow temperature gradient are improved by adding more snow layers. However, we understand the reviewer's concerns and experiences, and the minimum threshold of the number of snow layers suggested by existing studies ranges from 3 to 20 depending on locations, time periods, and model setup. Therefore, we tone down the sentences and provided more citations in the revised manuscript. The related sentences included in the revised manuscript are presented in the previous comments)

**\* References**
Aoki, T., Kuchiki, K., Niwano, M., Kodama, Y., Hosaka, M., & Tanaka, T.: Physically based snow albedo model for calculating broadband albedos and the solar heating profile in snowpack for general circulation models. Journal of Geophysical Research, 116, https://doi.org/10.1029/2010JD015507, 2011.
Augas, J., Abbasnezhadi, K., Rousseau, A.N., & Baraer, M.: What is the Trade-Off between Snowpack Stratification and Simulated Snow Water Equivalent in a Physically-Based Snow Model?. Water, 12, 3449, https://doi.org/10.3390/w12123449, 2020.
Cheng, B., Zhang, Z., Vihma, T., Johansson, M., Bian, L., Li, Z., & Huiding, W.: Model experiments on snow and ice thermodynamics in the Arctic Ocean with CHINARE 2003 data. Journal of Geophysical Research, 113(C9), C09020, https://doi.org/10.1029/2007JC004654, 2008.
Jennings, K.S., Kittel, T.G.G., & Molotch, N.P.: Observations and simulations of the seasonal evolution of snowpack cold content and its relation to snowmelt and the snowpack energy budget. The Cryosphere, 12, 1595–1614, https://doi.org/10.5194/tc-12-1595-2018, 2018.
Lecomte, O., Fichefet, T., Vancoppenolle, M., & Nicolaus, M.: A new snow thermodynamic scheme for large-scale sea-ice models. Annals of Glaciology, 52(57), 337–346, https://doi.org/10.3189/172756411795931453, 2011.

Livneh, B., Xia, Y., Mitchell, K.E., Ek, M.B., & Lettenmaier, D.P.: Noah LSM snow model diagnostics and enhancement. Journal of Hydrometeorology, 11, 721–738, https://doi.org/10.1175/2009JHM1174.1, 2010.

Saha, S.K., Sujith, K., Pokhrel, S., Chaudhari, H.S., & Hazra, A.: Effects of multilayer snow scheme on the simulation of snow: Offline Noah and coupled with NCEP CFSv2. Journal of Advances in Modeling Earth Systems, 9, 271–290, https://doi.org/10.1002/2016MS000845.

**Line 344 – 345:** "However, as proven in our findings, effective uses of the gamma SWE (e.g. localization function) will maximize the utility of the gamma SWE into the DA framework." I would scale back the certainty in this statement: e.g., replace "proven in" with "supported by" and "will maximize" with "can enhance."

Thank you for the suggestions. We edited them as suggested.

"However, as supported by our findings, effective uses of the gamma SWE (e.g. localization function) can enhance the utility of the gamma SWE into the DA framework."

**Lines 347- 348:** "promising alternative"; Alternative to what?

We changed this to "approach"

"DA has been used as a promising approach to improve SWE estimation at a large spatial scale"

**Lines 356 – 358:** "The added value of the gamma data on the model SWE estimates was greater for the relatively lower VCF range. For areas with higher topographic heterogeneity, the gamma-based DA SWE was still effective in reducing the errors."

This is very awkwardly worded, the first sentence is in reference to the vegetation fraction, and the second was for the topography heterogeneity. Specifically, the use of the phrase "still effective" is confusing since there is no "base-state" effectiveness of the DA for terrain heterogeneity described here in the conclusions. Please reword.

We agreed with the Reviewer's comment. The second sentence was reworded as below.

"While the gamma-based DA SWE had relatively lower improvement in areas with higher topographic heterogeneity, the DA SWE with reduced errors was found as compared to the OL.

**Lines 361 – 362:** "uncertainties in the Noah-MP physics (i.g., precipitation partitioning and simplified snow layers)." I really do not think that this statement belongs in the conclusions, since it's just speculation on the part of the authors.

We removed this part in the conclusion.

**Data Availability:**

Definitely a lot to be desired here, by only pointing readers to publicly available datasets, and choosing not to include model configuration data, analysis code, or processed (example) data, there is simply no way that a reasonable person would be able to repeat the experiment and analysis described in the paper. At a bare-minimum, the model configuration files should be included, as should any code that processes the data prior to being fed into the model.

As we answered ablow, we will provide all lis.config files used to run the simulations in supplemental materials. And original and reformatted airborne gamma SWE data (as well as the R code used to

reformat them) as well as time series outputs from the OL and DA runs (as much as the size is allowed) will be  available for download at [will add a link to data from Zenodo, currently being setup with an ODC Attribution (ODC-BY) license for access without restrictions].

**Author Contributions:**
I'll leave this to the editors, but I'm not sure "provided the funding" is a good reason to include for co-authorship. While in this instance it doesn't matter since SVK and CMV provided other input that would merit co-authorship, explicitly listing "provided funding" seems like it could be problematic.

Thank you for the comment. We removed the listing "Acquired the funding and the resources" here. As we stated, SVK and CMV helped the formal analysis, particularly localization DA implements in LIS, provided technical and scientific throughout a series of meetings and discussions, and actively reviewed and edited the manuscript.

---

## Author Comment (AC2)

**Reviewer 2**
Review of " Assimilation of airborne gamma observations provide utility for snow estimation in forested environments" by Cho et al.

SUMMARY:
Overall, I think this paper is relevant for publication in HESS with a clear presentation. This paper leverages airborne gamma observations to estimate snow water equivalent using data assimilation. This is great work for the snow community that shows the potential of remote-sensed gamma observations to improve snow estimates. That said, clarification of how localization is implemented with DA would be helpful for others to repeat your work and interpolate the results.

Thank you for your positive feedback and the valuable comments on our manuscript. We have carefully revised our manuscript based on your comments.

Line-by-line comments:

1. L125: For Equations (1) to (4), make sure to use the same format for the uncollided gamma count rates (e.g., $40K_b$ $^{40}K_b$).

Thank you for your keen eye. We edited this.

2. L138: From my understanding, COOP snow depth is first converted to SWE which is assimilated (rather than snow depth) to get UA SWE. Please make sure this sentence won't cause any confusion.

We agreed with the comment. The sentence was edited to clarify the data development procedure as below.

*"The UA SWE is the ground observation-based 4-km gridded SWE product developed by consistently assimilating the snow telemetry (SNOTEL) SWE and NWS Cooperative Observer Program (COOP) snow depth measurements (which was first converted to SWE using a newly developed snow density parameterization) with the Parameter-elevation Regressions on Independent Slopes Model (PRISM) temperature and precipitation data over the continental United States"*

3. L186: For MERRA2, are bias-corrected precipitation or uncorrected precipitation used as inputs? In line 308, the authors mention that overestimated SWE is likely attributed to precipitation phase partition. Would bias from precipitation contribute to the overestimation?

We used the uncorrected MERRA-2 forcing in the original manuscript because we focused on demonstrating the feasibility of the gamma SWE DA to improve the model estimates of SWE, particularly in forested areas, using the atmospheric forcing as is (i.e., without bias-correction), which is a typical case of operational prediction or monitoring systems. However, in order to analyze the effect of the atmospheric forcing on the snow accumulation, we conducted additionally experiments using the bias-corrected MERRA-2 forcing (as well as different parameterization schemes to address another Reviewer's comments). The figure below shows that the use of bias-corrected MERRA-2 forcing was

effective in improving the SWE estimates during the snow accumulation period as compared to the case using the original (i.e., uncorrected) MERRA-2 forcing. Here, it is also worth to note that assimilation of the gamma SWE data (using the uncorrected MERRA-2 forcing) provides similar SWE estimates to the case of using the bias-corrected forcing when the gamma SWE observations are available during the snow accumulation period.

We included these analyses in the revised manuscript as follows:

*"These issues can be attributed to parameterization schemes and/or atmospheric forcing employed in Noah-MP. Parameterization options for the precipitation phase partitioning method, ground surface albedo, surface layer drag coefficient, and snow/soil temperature time scheme can affect the snow simulations (You et al., 2020). To further analyze the issues, we conducted additional experiments using different parameterization schemes and atmospheric forcing. That is, the BATS scheme for partitioning precipitation into rainfall and snowfall, CLASS scheme for ground surface albedo, Chen97 scheme for surface layer drag coefficient, fully-implicit snow and soil temperature time scheme, and the bias-corrected MERRA-2 forcing were additionally tested. As shown in **Figure 8a**, the use of BATS or CLASS schemes do not make a significant difference in the SWE estimates as compared to the original OL results. Although the Chen97 or fully-implicit schemes are effective in delaying the snow removal date, they add considerably more snow during the snow accumulation period and do not help capture snowmelt start date (**Figure 8a**). Furthermore, the effectiveness of each parameterization scheme varies with flight lines and time periods within the study domain as also emphasized by You et al. (2020).*

*Figure 8b shows that the use of the bias-corrected MERRA-2 forcing is effective in improving the SWE estimates during the snow accumulation period, but it still has the issue of rapid snow melting. The combinational use of the bias-corrected MERRA-2 forcing and fully-implicit scheme leads to improved snow removal timing, but largely overestimated SWE during the snow accumulation period. We originally used the uncorrected MERRA-2 forcing to demonstrate the feasibility of the gamma SWE DA for improving the model estimates of SWE, particularly in forested areas, using the atmospheric forcing as is (i.e., without bias-correction), which is a typical case of operational prediction or monitoring systems. Here, it is worth to note that assimilation of the gamma SWE data provides similar SWE estimates to the case of using the bias-corrected forcing with semi-implicit scheme when the gamma SWE observations are available during the snow accumulation period."*

[Figure]

**Figure 8**. Examples of SWE time series including (a) the additional open-loop (OL) experiments using different parameterization schemes related to snow simulations such as the snow/soil temperature time scheme (semi-implicit vs. fully-implicit), partitioning precipitation phase (Jordan91 vs. BATS), ground surface albedo (BATS vs. CLASS), and surface layer drag coefficient (Monin-Obukhov [M-O] similarity theory vs. Chen97 [original Noah]), and (b) DA runs forced by original vs. bias-corrected MERRA2 forcings with each snow/soil temperature time scheme which is a parameterization option largely affecting snow simulations.

4. L196: Could you briefly summarize how Kwon et al. (2021) perturbs the forcings? For example, which forcings are perturbed? How are the parameters chosen? (i.e., Were in situ observations used to

quantify these parameters?) This would be helpful to know if precipitation uncertainties are considered.

Perturbation parameters applied during the OL and DA runs are summarized in the following table, which was included in the revised manuscript. The perturbation parameter values in Kwon et al. (2021) were determined based on previous DA studies (e.g., Forman et al., 2012; Kumar et al., 2009; 2014; 2016; Reichle et al., 2008).

**Table 1**. Perturbation parameters applied to model prognostic state variables and atmospheric forcing fields during the OL and DA runs.

| Variable | Perturbation Types | Std dev | AR(1) | Cross correlations | | |
|---|---|---|---|---|---|---|
| Model prognostic state variables | | | | $SWE$ | $SD$ | |
| $SWE$ | M | 0.01 | 3 hr | – | 0.9 | |
| Snow depth ($SD$) | M | 0.02 | 3 hr | 0.9 | – | |
| Atmospheric forcing fields | | | | $SW$ | $LW$ | $P$ |
| Shortwave radiation ($SW$) | M | 0.3 | 1 day | – | -0.5 | -0.8 |
| Longwave radiation ($LW$) | A | 50 W m$^{-2}$ | 1 day | -0.5 | – | 0.5 |
| Precipitation ($P$) | M | 0.5 | 1 day | -0.8 | 0.5 | – |

M: multiplicative; A: additive; AR(1): first-order autoregressive temporal correlation.

**\* References**

Forman, B.A., Reichle, R.H., & Rodell, M.: Assimilation of terrestrial water storage from GRACE in a snow-dominated basin. Water Resources Research, 48(1), W01507, https://doi.org/10.1029/2011WR011239, 2012.

Kumar, S.V., Reichle, R.H., Koster, R.D., Crow, W.T., & Peters-Lidard, C.D.: Role of subsurface physics in the assimilation of surface soil moisture observations. Journal of Hydrometeorology, 10 (6), 1534–1547, https://doi.org/10.1175/2009JHM1134.1, 2009.

Kumar, S.V., Peters-Lidard, C.D., Mocko, D., Reichle, R., Liu, Y., Arsenault, K.R., Xia, Y., Ek, M., Riggs, G., Livneh, B., & Cosh, M.: Assimilation of remotely sensed soil moisture and snow depth retrievals for drought estimation. Journal of Hydrometeorology, 15 (6), 2446–2469, https://doi.org/10.1175/JHM-D-13-0132.1, 2014.

Kumar, S.V., Zaitchik, B.F., Peters-Lidard, C.D., Rodell, M., Reichle, R., Li, B., Jasinski, M., Mocko, D., Getirana, A., De Lannoy, G., Cosh, M.H., Hain, C.R., Anderson, M., Arsenault, K.R., Xia, Y., & Ek, M.: Assimilation of gridded GRACE terrestrial water storage estimates in the North American Land Data Assimilation System. Journal of Hydrometeorology, 17 (7), 1951–1972. https://doi.org/10.1175/JHM-D-15-0157.1, 2016.

Reichle, R.H., Crow, W.T., & Keppenne, C.L.: An adaptive ensemble Kalman filter for soil moisture data assimilation. Water Resources Research, 44 (3), W03423, https://doi.org/10.1029/2007WR006357, 2008.

5. L215: It is not clear to me how localization is applied in the assimilation. Does it weigh the covariance matrix? It might be better to link equation (7) with the relevant equation mentioned above.

I might not fully understand it, but why localization used to update SWE estimates would impact the open loop results shown in figure 6? I assume localization would only impact DA SWE.

The localization is applied in the assimilation by weighting distances from the flight lines (up to a specified localization distance; *r*) using the Gaussian decay-based function. The degree of the SWE updates for a grid cell from the assimilation is calculated using a localization weight (W) which is calculated based on the distance (*d*) from the updated grid cells overlapped with the flight line.

Yes, the localization only impacts DA SWE. The reason why the OL statistics were changed with increasing the localization distances in Figure 6 is that the effective distance used to calculate the domain-averaged OL SWE was changed according to the localization distance. For example, at a localization distance of 4 km, the domain-averaged OL SWE is calculated using gridded SWE values within an effective surrounding area (4 km distance from the gamma flight lines). At 32 km, the domain-averaged OL SWE was calculated based on larger number of the gridded SWE values within surrounding areas up to 32-km distance from the gamma flight line. The OL/DA statistics in Figure 6 are calculated using domain-averaged time series of OL/DA SWE over the effective surrounding areas by localization distances with the corresponding UA SWE.

We edited and added descriptions in Section 4.3 as below.

*"To quantify if the spatially sparse gamma SWE observations can improve the SWE estimates in the surrounding areas, where the observations are not available, we apply a distance-based localization method into the assimilation procedure. The localization is applied in the assimilation by weighting distances from the flight lines (up to a specified localization distance; r) using the Gaussian decay-based localization method as follows:*

$$W = exp\left\{\frac{-d^2}{2\cdot\left(\frac{r}{2}\right)^2}\right\} \tag{7}$$

*where d is the distance between the updated grid cells (i.e., flight lines) and grid cells without observations within a specified localization radius r. The degree of the SWE updates for a grid cell from the assimilation is calculated using the localization weight (W) which is calculated based on the distance (d) from the updated grid cells overlapped with the flight line."*

When double-checking our calculation for Figure 6, we found a script error when calculating the error metrics. The error metrics were miscalculated using a "static" domain-averaged OL SWE time series with DA SWE time series at different localization distances. They should have been calculated with a domain-averaged OL SWE time series at a "corresponding" localization distance. We reviewed throughout the script, corrected the part, and recalculated the metrics which are shown in the new figure below.

[Figure]

**Figure 6**. Localized data assimilation (DA) and open-loop (OL) Noah-MP SWE performances as compared to the UA SWE with different localization distances (e.g., 4, 8, 16, 24, 32, and 48 km) for the whole (accumulation and melting periods), accumulation, and melting periods, respectively.

The result parts of the manuscript related to the revised figure 6 were also revised as below.

*"For the whole snow season that includes both accumulation and melting periods, the boxplot of the 1:1 slope shows that the localized DA SWE were improved as compared to OL. The slopes of the DA SWE are closer to 1 than the OL's slopes. The bias and RMSD boxplots also show that the DA SWE has lower errors than the OL SWE for all localization distances, expect for bias at 48 km which is too low (median: – 23 mm). The OL's RMSDs slightly increased at the distances up to 16 km (median: 72 mm) and decreased after that, while the DA's RMSD values continually decreased with increasing the distances up to 48 km (median: 53 mm). When the statistics were calculated for the accumulation and melting periods separately, the lower RMSDs and slopes closer to 1 of the localized DA SWE were found consistently. As previously discussed, the efficacy of assimilating the airborne gamma SWE is greater during the*

*accumulation period, especially for bias and RMSD, than during the melting period. In the melting period, the improvements in the bias, RMSD and 1:1 slope are also achieved up to 32 km"*

6. L226: maybe use lowercase "A" in the parenthesis.

Edited.

7. L236: it seems peak SWE might not be correctly estimated if only one data point exists after the accumulation season (Figure 4 SJ150 in WY 1989 and NH106 in WY 1997). It might be worth pointing that out and/or discussing this issue.

We agreed with your comment. To point out the issue, we added a statement as below.
"Also, the peak SWE cannot be corrected if a single gamma SWE exists only after the accumulation period."

8. Please be consistent with either RMSD or RMSE throughout the manuscript.

We edited and consistently used RMSD throughout the manuscript.

---

## Referee Report (RR1)

TITLE: " Assimilation of airborne gamma observations provides utility for snow estimation in forested environments" by Cho et al.

SUMMARY AND RECOMMENDATION

The authors have addressed the comments I raised. In this revision, both bias-corrected and uncorrelated MERRA2 precipitation are used for comparison. The authors also tested three parameterization schemes. The results show bias-corrected MERRA2 forcings and fully-implicit would improve SWE snow removal timing. Additionally, the author also provided more detailed about the localization. The provided information is useful for others to repeat the results. I would suggest accepting this revision as is.

---

## Referee Report (RR2)

**Second Review of:** Assimilation of airborne gamma observations provides utility for snow estimation in forested environments

**Overview:** The revised manuscript has adequately addressed a majority of my major concerns during the first round of review, including performing some simple sensitivity analysis with Noah-MP to address the poor performance of the open loop simulation, and including some additional details regarding model set up and methods as well as significant steps to improve replicability of the study. Furthermore, the DA of gamma flightline measured SWE in forested regions of the Northeast US is of potential value since this can address a number of issues in the region related to snow characterization on fine scales. In particular, the result quantifying the impacts of localization distance on model performance can inform future data collection strategies and constrain regional SWE estimates from blended model/observational approaches. The DA approach is reasonable, and the gamma-SWE dataset is well validated and widely accepted within the community. Taken altogether, this study is of potential high-value to the community and worthy of publication. However, there are still some lingering larger concerns with the study that should be addressed prior to publication. Additionally, the modifications during the first round of revisions introduced a number of minor technical issues that need to be corrected.

**Major comments:**
1. While the authors have made substantial strides towards addressing the poor performance of the OL simulations, it is still concerning just how bad the model performance appears to be. While, there are and can be fairly large model errors with Noah-MP, particularly around SWE max and during the melt season these are by far the most egregious that I can recall seeing in the literature. Accordingly, because the results of the OL simulation are so questionable for a widely used and accepted numerical model, I think the bar to publishing these results should be quite high. In the first revision, the authors have made solid efforts to add context to the model performance, however to reach this bar, in addition to what the authors have already done, I recommend adding (or at least investigating, even if these results don't end up in the final manuscript) three specific things:

    a. Find a single example location within the model domain where with an in-situ SWE measurement and compare SWE from the UA dataset, the OL simulation, and different model-based product for snow (e.g., NLDAS). Snow depth could be used as a backup if there are no in-situ SWE measurements in the model domain.
    b. Perform a simple "reality check" analysis of the MERRA-2 forcing against a handful (or even a single) in-situ weather station to explore possible biases associated with the reanalysis forcing in a more direct way than comparing LSM results from different versions.
    c. Include one more Noah-MP simulation with the rain/snow partitioning threshold set to 0.0 instead of using the BATS = 2.5C threshold. I suspect that the lack of sensitivity to the precipitation phase partitioning is tied to the fact that both Jordan (1991) and BATS have that 2.5 threshold for rain/snow within them.

Addressing these three items would do the following: 1) directly contextualize and ground truth the OL model results and the UA dataset with an on-the-ground SWE observation within the region, 2) illustrate, directly, possible biases in the model forcing, and 3) provide a full sensitivity analysis to phase-partitioning. Once these have been done, I think that frees up the authors to be more speculative regarding the model performance, and perhaps even make more general statements on how to improve the model in this region.

2. To me there appears to be some weirdness going on in figure 4 that should be explained.
    a. How does the AMSR-E DA run end up with a later melt out date than all other simulations despite consistently lower SWE throughout the season, and a number of "zero" SWE assimilations in the spring? That is, why is the AMSR-E snowmelt rate dramatically more gradual than the OL or the Gamma-DA simulations?
    b. What is going on at the Rumney time-series that allows for a late-season spikes in SWE time-series in the Gamma-DA simulation that is not reflected in the OL simulation? My first assumption was that there was a late-season snow event each year, but then wouldn't that also show up in the OL simulation? I'm confused how there is this spike in SWE in the DA simulation that appears without a flightline gamma observation to explain it.

3. Additional model details would be helpful here in section 4. For example, if the model is run over a gridded domain, what is the grid-spacing? How does this grid-spacing compare to the MERRA forcing? Would it be helpful to show an outline of the model domain in figure 1? Is figure 1 already *showing* the model domain? Finally, while the authors indicated in their responses that the MERRA-2 data was interpolated to the LIS grid, was it *downscaled* to the terrain at all? (e.g., was temperature adjusted using a lapse-rate?). There is a lot of terrain in NH and ME that the flightlines sample, and the MERRA forcing almost certainly doesn't capture it adequately.

**Minor comments:**

Line 137: What does UA stand for, I think this is the first time this acronym shows up, please define it.

Line 139: What is the snow density parameterization, reference?

Line 191: The BATS and CLASS albedo schemes are specifically for the snow albedo, not the total ground albedo, please correct this.

Line 197: Would it be helpful to elaborate on the purpose of the ensemble here, e.g., to generate model uncertainty metrics for the DA?

Line 224: "of limited used" should be "of limited use"

Line 231: Consider replacing "the degree of the SWE updates" with "the magnitude of the SWE DA adjustment" or something along those lines.

Line 244 – 249: I suggest rewording "However, this was a consequence of the fact that the overestimated SWE during the accumulation season and early in the melt season was offset by the underestimated SWE during the snowmelt season (i.e., April and May). When the gamma SWE observations exist during the accumulation period (which is a typical case), DA corrected the overestimated SWE, whereas it further underestimated SWE in the snowmelt season (**Figures 3** and **4**), resulting in the increased (negative) bias, as presented in **Figure 2**."

As

*However, this was a consequence of the fact that in correcting the overestimated SWE during the accumulation season, the DA introduced a greater underestimate during the melt season (Figures 3 and 4).*

Line 251:  I suspect that the r-value decrease is due almost entirely to the increase in the number of zero – to – not-zero comparisons involved in the analysis.  The physical lower-boundary of snow as a variable really complicates a lot of these statistical metrics.  Perhaps a BETTER comparison would be to only compare data-points where both the simulated and observed SWE are greater than zero.

Line 261: Again, I have trouble with the statement "limited model physics" here. Pending changes made to address major comment 1 above, I think this can be addressed with a simple change of wording to something along the lines of (changes in ***bold/italics***):

"However, the assimilation of the airborne gamma SWE measurements was not able to improve the snow ablation timing ***due to sparse gamma data during the spring in combination with the overall poor model performance during the melt season***"

Essentially, any rewording that more generally acknowledges the poor performance in this specific instance, rather than speculating that there is something intrinsically wrong with the model physics.

Line 264: "single gamma SWE" should be "single gamma SWE flight?"

Line 273: I suggest that you remove "In the figure" from the beginning of the sentence.

Line 275: "has low bias and RMSD than OL SWE". This is an incomplete though, the model has a low bias, and a ??? RMSD compared to OL SWE? Higher? Lower? Equivalent? Please fix.

Line 275: Suggested replace: "DA performances show relatively" to "DA led to"

Line 279 (and more generally throughout the manuscript): Consider replacing the word "updated" with "improved." Using the word updated is ambiguous regarding whether or not the simulation was improved. I'll note it here, but I recall seeing the use of "updated" where a better word choice is possible at other places throughout the manuscript as well.

Line 294: consider replacing "surrounding areas, where gamma flights do not exist" with "In areas surrounding the gamma-flights"

Figure 6 and Lines: 297 – 307: This is a nice result. Is there a way to assess statistical significance in this comparison? E.g., at what localization distance are the improvements compared to the OL no longer statistically significant (or are all of these significant)? The results here are somewhat convincing visually, so I leave any decision to include a statistical significance analysis here to the authors.

Line 297: what is the "effective surrounding areas"? is this a fixed number for ALL localization radii? Or only gridcells within the localization radii?

Line 301: Why is the OL RMSD decreasing as localization radii?

Line 316: replace "reduced the model SWE errors" with "improved model SWE"

Line 317: There is an errant open parenthesis here.

Line 332: replace the word "ground" with the word "snow"

Line 334: add, "snow albedo" after "BATS and CLASS"

Line 340: replace "but it still has" with "but did not improve"

Line 340: replace "combinational" with "combined"

Line 341: "fully-implicit" should be "the fully-implicit"

Line 344: replace "worth to note" with "worth noting"

Line 364: I don't think I expect much of a different between Jordan and BATS precipitation, but rather differences to be most pronounced when T thresh is set to 0C (Letcher et al. 2022 explores the 2.5 vs. 0C difference in this region).

Lines 367 and 370, please replace the word "Noah-MP" with "BATS" since Noah-MP has different options for precipitation phase partitioning, so the 2.5C threshold is specific to the BATS choice, not the Noah-MP model.

---

## Author Response (AR2)

Dear authors,

The revised manuscript has adequately addressed a majority of the concerns from two reviewers during the first round of review. However, as you can see, there are still some issues which need to be further addressed, especially the specific comments from Anonymous referee #1. Hence, this MS is still subject to revisions before considering for publication.

Best regards,
Hongkai Gao

[Answer]
Dear Dr. Hongkai Gao,

We really appreciate your time handling our manuscript. Based on the Reviewer #1 comments, we carefully revised the manuscript including additional results of SWE comparison between Noah-MP runs and in-situ observations at a snow pillow site (Hubbard Brook, New Hampshire. We also re-ran the AMSR2 DA simulations to correct errors and revised Figures 4 & 7 with additional descriptions. For further details, please see our responses to relevant comments given by Reviewer #1.
   We consider this modified version of the manuscript to be significantly improved because of those comments, and therefore hope that this revision will sufficiently address the reviewer's concerns. Thank you again for your time and efforts.

Sincerely,
Eunsang Cho, Yonghwan Kwon, Sujay Kumar, and Carrie Vuyovich

**Second Review of:** Assimilation of airborne gamma observations provides utility for snow estimation in forested environments

**Overview:** The revised manuscript has adequately addressed a majority of my major concerns during the first round of review, including performing some simple sensitivity analysis with Noah-MP to address the poor performance of the open loop simulation, and including some additional details regarding model set up and methods as well as significant steps to improve replicability of the study. Furthermore, the DA of gamma flightline measured SWE in forested regions of the Northeast US is of potential value since this can address a number of issues in the region related to snow characterization on fine scales. In particular, the result quantifying the impacts of localization distance on model performance can inform future data collection strategies and constrain regional SWE estimates from blended model/observational approaches. The DA approach is reasonable, and the gamma-SWE dataset is well validated and widely accepted within the community. Taken altogether, this study is of potential high-value to the community and worthy of publication. However, there are still some lingering larger concerns with the study that should be addressed prior to publication. Additionally, the modifications during the first round of revisions introduced a number of minor technical issues that need to be corrected.

[Answer] We very much appreciate the Reviewer's time and valuable comments. We included additional results with in-situ SWE observations and another run with the NLDAS2 forcing data. We also addressed the reasons for showing some weird patterns in the SWE time series and re-ran the AMSR2 DA simulations with unit corrections. We hope that this revision will sufficiently address the Reviewer's concerns. Please see the details below.

**Major comments:**

**1.** While the authors have made substantial strides towards addressing the poor performance of the OL simulations, it is still concerning just how bad the model performance appears to be. While there are and can be fairly large model errors with Noah-MP, particularly around SWE max and during the melt season these are by far the most egregious that I can recall seeing in the literature. Accordingly, because the results of the OL simulation are so questionable for a widely used and accepted numerical model, I think the bar to publishing these results should be quite high. In the first revision, the authors have made solid efforts to add context to the model performance, however, to reach this bar, in addition to what the authors have already done, I recommend adding (or at least investigating, even if these results don't end up in the final manuscript) three specific things:

**a.** Find a single example location within the model domain where with an in-situ SWE measurement and compare SWE from the UA dataset, the OL simulation, and different model based product for snow (e.g., NLDAS). Snow depth could be used as a backup if there are no in-situ SWE measurements in the model domain.

[Answer] According to the Reviewer's suggestion, we found a SNOTEL/SCAN site at Hubbard Brook, NH where in-situ snow pillow SWE measurements are available, and made an SWE time series plot from Oct 1, 2002, to May 31, 2003, along with the UA observations, OL (MERRA2), and a new Noah-MP simulation forced by NLDAS2 forcing data. As we found previously, the OL-MERRA2 simulation overestimated SWE while the SWE simulations from NLDAS2 with the same Noah-MP parameterization options were close to the in-situ and UA SWE, which supports our previous analysis (in the first round of revision) that meteorological forcing is the main reason for the large overestimation of SWE during the accumulation period. However, the pattern of the rapid snow melting is also seen in the model results driven by NLDAS2.

[Figure]

**SNOTEL − Hubbard Brook, NH**

**b.** Perform a simple "reality check" analysis of the MERRA-2 forcing against a handful (or even a single) in-situ weather station to explore possible biases associated with the reanalysis forcing in a more direct way than comparing LSM results from different versions.

[Answer] Thank you for the suggestion. To address this, we compared the MERRA2 forcing precipitation to the in-situ precipitation along with the NLDAS2 forcing precipitation below. The figure showed that both reanalysis forcings substantially overestimated precipitation between November and December. This overestimated precipitation led to overestimated SWE simulations particularly with MERRA-2.

[Figure]

**c.** Include one more Noah-MP simulation with the rain/snow partitioning threshold set to 0.0 instead of using the BATS = 2.5C threshold. I suspect that the lack of sensitivity to the precipitation phase partitioning is tied to the fact that both Jordan (1991) and BATS have that 2.5 threshold for rain/snow within them.

[Answer] To address the Reviewer's concern, we ran two more Noah-MP simulations with the rain/snow partitioning threshold set to 0.0 C with MERRA2 and NLDAS2, respectively. The figure below shows the SWE differences between Jordan 1991 (2.5 C) and 0.0 C. Based on the results, a change in the precipitation phase partitioning threshold from Jordan 1991 (2.5 C) to 0.0 C led to the SWE decrease up to 65 mm for MERRA2 and 20 mm for NLDAS2. This also confirms that the MERRA-2 forcing is responsible for the large overestimation of SWE rather than the parameterization option, which although contributes to a small portion of the SWE overestimation.

[Figure]

**Figure 9**. Comparison of SWE time series between four Noah-MP simulations and the Soil Climate Analysis Network (SCAN) ground-based observations at Hubbard Brook, New Hampshire from Oct 1, 2002, to May 31, 2003 (https://wcc.sc.egov.usda.gov/nwcc/site?sitenum=2069). The four Noah-MP SWE simulations were generated with Jordan (1991)'s scheme and a single threshold of 0°C and two meteorological forcings (MERRA2 – which is used for OL and the North American Land Data Assimilation System; NLDAS2), respectively.

Addressing these three items would do the following: 1) directly contextualize and ground truth the OL model results and the UA dataset with an on-the-ground SWE observation within the region, 2) illustrate, directly, possible biases in the model forcing, and 3) provide a full sensitivity analysis to phase-partitioning. Once these have been done, I think that frees up the authors to be more speculative regarding the model performance, and perhaps even make more general statements on how to improve the model in this region.

[Answer] Again, thank you for the valuable suggestions. Through these items, we found that the overestimated SWE simulations dominantly resulted from the reanalysis of meteorological forcings (precipitation) rather than the Noah-MP model itself.

We include the last figure in the revised manuscript (Figure 9) with descriptions as below.

"Letcher et al. (2021) demonstrated that the use of cooler Tair thresholds in Noah-MP can improve the estimates of peak SWE in the northeastern United States. To verify this, four Noah-MP SWE simulations with Jordan (1991)'s scheme and a single threshold of 0°C with two different meteorological forcings (MERRA2 and the North American Land Data Assimilation System; NLDAS2) are compared to ground-based SWE observations from Oct 1, 2002, to May 31, 2003, at Hubbard Brook, New Hampshire, which is within the study domain (Figure 9). This supports the previous finding that the overestimated SWE with Jordan's scheme was reduced with a single threshold of 0°C for both forcings. This also presents that the use of regionally reliable meteorological forcings (e.g., precipitation) generates accurate SWE estimations.

2. To me there appears to be some weirdness going on in figure 4 that should be explained.

a. How does the AMSR-E DA run end up with a later melt out date than all other simulations despite consistently lower SWE throughout the season, and a number of "zero" SWE assimilations in the spring? That is, why is the AMSR-E snowmelt rate dramatically more gradual than the OL or the Gamma-DA simulations?

[Answer] We appreciate you for pointing this out. We carefully investigated the AMSR2 DA processes and found that there was a unit error when regridding the original AMSR2 SWE (10 km) to the OL grid (4 km) before the DA process. We originally thought the unit of AMSR2 SWE is "cm" so the SWE values were multiplied by 10 to make "mm" though the conversion was not needed as the original unit is "mm". This error caused the weird DA simulations with a later melt-out date than OL. We re-ran the AMSR2 DA simulations with the unit correction for all gamma lines and regenerated the time series (Figure 4) and boxplot (Figure 7). To help compare, the time series of the AMSR2 DA with/without the unit correction (SJ150) are provided below. The revised figures 4 & 7 are also attached.

[Figure]

[Figure]

**Figure 4**. Examples of daily SWE time series of three gamma lines (SJ150, NH106, and NH109) with latitude (Lat), longitude (Lon), elevation (Elev), and vegetation cover fraction (VCF) for individual years including the open-loop (OL) and gamma data assimilated (DA_Gamma) Noah-MP SWE estimates along with the passive microwave SWE data from the Advanced Microwave Scanning Radiometer 2 (AMSR2) and AMSR2 data assimilated SWE (DA AMSR2).

[Figure]

**Figure 7**. Comparison of the SWE estimation performance between the open-loop (OL), gamma DA, and AMSR2 DA as compared to the UA SWE at the 16km localization distance for the mutual DA effective accumulation periods.

b. What is going on at the Rumney time-series that allows for a late-season spikes in SWE time-series in the Gamma-DA simulation that is not reflected in the OL simulation? My first assumption was that there was a late-season snow event each year, but then wouldn't that also show up in the OL simulation? I'm confused how there is this spike in SWE in the DA simulation that appears without a flightline gamma observation to explain it.

[Answer] We carefully investigated the time series at Rumney, NH, and found that there was an issue when calculating a grid-average SWE time series for the NH109 gamma line. A pixel that should be excluded was included in the calculation, resulting in the late-season spikes. Here is a time series plot with grid-average SWE values before/after the grid correction. The grid correction also led to slight changes in the OL time series. We replaced it with the new time series for NH109 in Figure 4.

**Before** the grid correction

[Figure]

**After** the grid correction

3. Additional model details would be helpful here in section 4. For example, if the model is run over a gridded domain, what is the grid-spacing? How does this grid-spacing compare to the MERRA forcing? Would it be helpful to show an outline of the model domain in figure 1? Is figure 1 already *showing* the model domain? Finally, while the authors indicated in their responses that the MERRA-2 data was interpolated to the LIS grid, was it *downscaled* to the terrain at all? (e.g., was temperature adjusted using a lapse-rate?). There is a lot of terrain in NH and ME that the flightlines sample, and the MERRA forcing almost certainly doesn't capture it adequately.

[Answer] Thanks for the suggestion. The model was run over a gridded study domain (LAT: 42.76 to 47.44 / LON: -72.29 to -67.85; Figure 1) with a grid spacing of 0.04° while the spatial resolution of the MERRA-2 forcing is 0.625° (latitude) × 0.5° (longitude). Therefore, the original MERRA-2 forcing was downscaled to the model grid using the bilinear method within LIS. During the interpolation, topographic correction methods were applied for air temperature, air pressure, humidity, and longwave radiation based on the lapse rate (Cosgrove et al., 2003), and for longwave radiation by considering the impact of topographic slope and aspect (Dingman, 2002). These explanations were included in the revised manuscript.
**\* References**

Cosgrove, B.A., Lohmann, D., Mitchell, K.E., Houser, P.R., Wood, E.F., Schaake, J.C., Robock, A., Marshall, C., Sheffield, J., Duan, Q., Luo, L., Higgins, R.W., Pinker, R.T., Tarpley, J.D., Meng, J.: Real-time and retrospective forcing in the North American Land Data Assimilation System (NLDAS) project. *Journal of Geophysical Research*, 108(D22), 8842. https://doi.org/10.1029/2002JD003118, 2003.

Dingman, S.L.: Physical Hydrology (2nd ed.), Prentice-Hall, 2002.

**Minor comments:**

Line 137: What does UA stand for, I think this is the first time this acronym shows up, please define it.

[Answer] We defined the University of Arizona (UA) here.

Line 139: What is the snow density parameterization, reference?

[Answer] We added a related reference, Dawson et al. (2017).

Dawson, N., Broxton, P., & Zeng, X.: A new snow density parameterization for land data assimilation. Journal of Hydrometeorology, 18(1), 197–207. https://doi.org/10.1175/JHM-D-16-0166.1, 2017.

Line 191: The BATS and CLASS albedo schemes are specifically for the snow albedo, not the total ground albedo, please correct this.

[Answer] Thank you for pointing this out. We corrected this.

Line 197: Would it be helpful to elaborate on the purpose of the ensemble here, e.g., to generate model uncertainty metrics for the DA?

[Answer] Thank you for the suggestion. We added the purpose of the ensemble here.

Then, using a restart file generated in the first step, an additional 3-year spin-up, from 1 January 1981 to 1 March 1984, was conducted using 20 ensemble members to generate model uncertainty metrics for the DA.

Line 224: "of limited used" should be "of limited use"

[Answer] We used "can be limited to be used"

Line 231: Consider replacing "the degree of the SWE updates" with "the magnitude of the SWE DA adjustment" or something along those lines.

[Answer] We replaced this with the suggested words.

Line 244 – 249: I suggest rewording "However, this was a consequence of the fact that the overestimated SWE during the accumulation season and early in the melt season was offset by the underestimated SWE during the snowmelt season (i.e., April and May). When the gamma SWE observations exist during the accumulation period (which is a typical case), DA corrected the overestimated SWE, whereas it further underestimated SWE in the snowmelt season (**Figures 3** and **4**), resulting in the increased (negative) bias, as presented in **Figure 2**."

As
*However, this was a consequence of the fact that in correcting the overestimated SWE during the accumulation season, the DA introduced a greater underestimate during the*

*melt season (Figures 3 and 4).*
[Answer] We reworded this as the Reviewer suggested. Thank you.

Line 251: I suspect that the r-value decrease is due almost entirely to the increase in the number of zero – to – not-zero comparisons involved in the analysis. The physical lower-boundary of snow as a variable really complicates a lot of these statistical metrics. Perhaps a BETTER comparison would be to only compare data-points where both the simulated and observed SWE are greater than zero.

[Answer] Thank you for the suggestion. We had actually considered those comparisons in the analysis. However, we concluded that the results based on zero-to-not-zero comparisons would be more appropriate because a calculation using non-zero values only could generate misinterpretation by removing the portion of adjusted SWE during the early snowmelt though this may result in a better R-value (we also think this would make sense for consistency with other statistical metrics such as bias).

Line 261: Again, I have trouble with the statement "limited model physics" here. Pending changes made to address major comment 1 above, I think this can be addressed with a simple change of wording to something along the lines of (changes in ***bold/italics***):

"However, the assimilation of the airborne gamma SWE measurements was not able to improve the snow ablation timing ***due to sparse gamma data during the spring in combination with the overall poor model performance during the melt season***" Essentially, any rewording that more generally acknowledges the poor performance in this specific instance, rather than speculating that there is something intrinsically wrong with the model physics.

[Answer] We agreed with the point. We changed the original statement with the suggested one the Reviewer made.

Line 264: "single gamma SWE" should be "single gamma SWE flight?"

[Answer] Changed.

Line 273: I suggest that you remove "In the figure" from the beginning of the sentence.

[Answer] Agreed. We removed.

Line 275: "has low bias and RMSD than OL SWE". This is an incomplete though, the model has a low bias, and a ??? RMSD compared to OL SWE? Higher? Lower? Equivalent? Please fix.

[Answer] We fixed by adding "lower RMSD".

Line 275: Suggested replace: "DA performances show relatively" to "DA led to"

[Answer] Replaced.

Line 279 (and more generally throughout the manuscript): Consider replacing the word "updated" with "improved." Using the word updated is ambiguous regarding whether or not the

simulation was improved. I'll note it here, but I recall seeing the use of "updated" where a better word choice is possible at other places throughout the manuscript as well.

[Answer] Thank you for your suggestion. We agreed and revised the word "updated" with "improved" throughout the manuscript. In certain statements whether or not the DA simulation was improved, we remain the word.

Line 294: consider replacing "surrounding areas, where gamma flights do not exist" with "In areas surrounding the gamma-flights"

[Answer] Replaced. Thank you.

Figure 6 and Lines: 297 – 307: This is a nice result. Is there a way to assess statistical significance in this comparison? E.g., at what localization distance are the improvements compared to the OL no longer statistically significant (or are all of these significant)? The results here are somewhat convincing visually, so I leave any decision to include a statistical significance analysis here to the authors.

[Answer] We agreed that Figure 6 here is sufficient enough to determine the magnitude of the improvements of the DA with different localization distances. We remain the current version with no inclusion of the significant test results.

Line 297: what is the "effective surrounding areas"? is this a fixed number for ALL localization radii? Or only gridcells within the localization radii?

[Answer] This means grid cells within the localization radii. We edited this statement as below.

"The OL/DA statistics in the figure are calculated using domain-averaged time series of OL/DA SWE for grid cells within a given localization distance with the corresponding UA SWE."

Line 301: Why is the OL RMSD decreasing as localization radii?

[Answer] This is because the spatial variability of SWE was smoothed and the domain-average values became similar, the domain-average OL and UA SWE for larger localization radii tend to have a lower RMSD.

Line 316: replace "reduced the model SWE errors" with "improved model SWE"

[Answer] Done.

Line 317: There is an errant open parenthesis here.

[Answer] We removed this.

Line 332: replace the word "ground" with the word "snow"

[Answer] Done.

Line 334: add, "snow albedo" after "BATS and CLASS"

[Answer] Added.

Line 340: replace "but it still has" with "but did not improve"

[Answer] Done.

Line 340: replace "combinational" with "combined"

[Answer] Done.

Line 341: "fully-implicit" should be "the fully-implicit"

[Answer] We edited it.

Line 344: replace "worth to note" with "worth noting"

[Answer] Replaced. Thank you for the detailed corrections.

---

## Author Response (AR3)

Dear authors,

Based on two reviewers' recommendations, they both agree that the science part is ready for publication. However, you can see the comments from Anonymous referee #1, that further correction of any remaining technical errors is suggested. Please read through the paper carefully again, to correct any grammatical or technical errors. Thus, I'd like to recommend "Publish subject to technical corrections".

Best regards,
Hongkai Gao

[Answer]
Dear Dr. Hongkai Gao,

We really appreciate your time handling our manuscript. Based on the Reviewer #1 comments, we carefully read through the manuscript, corrected several grammatical errors/typos, and added references we missed. Thank you again for your time and efforts.

Sincerely,
Eunsang Cho, Yonghwan Kwon, Sujay Kumar, and Carrie Vuyovich

**Reviewer 1**

**Final Comment of:** Assimilation of airborne gamma observations provides utility for snow estimation in forested environments

In this latest revision, the authors have sufficiently addressed my remaining concerns and made the corrections accordingly in the manuscript. In particular, I applaud their efforts in tracking down some of the sources of uncertainty in the forcing data and the model with the open loop simulation. This new analysis really helps contextualize the value of the DA and provides valuable insight into snow model performance in the region. Further, a final review of the paper didn't reveal and new or significant grammatical or technical errors that I could see. So, my recommendation is to accept and publish the manuscript in its current form. However, I always recommend a final read through by the authors to find and correct any remaining technical errors that might have been missed during the review process. Thank you, I look forward to seeing the publication in its final form.

[Answer]

We really appreciate your time providing valuable comments on our manuscript. We carefully did a final review of the recent version of the manuscript and corrected several grammatical errors with references.